

# Isoprenoid emission response to changing light conditions of English oak, European beech and Norway spruce

Ylva van Meeningen[1], Guy Schurgers[2], Riikka Rinnan[3] and Thomas Holst[1,3]

[1]Department of Physical Geography and Ecosystem Science, Lund University, Sölvegatan 12, 223 62 Lund, Sweden
[2]Department of Geosciences and Natural Resource Management, University of Copenhagen, Øster Voldgade 10, DK-1350 Copenhagen K, Denmark
[3]Terrestrial Ecology Section, Department of Biology, University of Copenhagen, Universitetsparken 15, 2100 Copenhagen E, Denmark

*Correspondence to*: Ylva van Meeningen, (ylva.van_meeningen@nateko.lu.se)

**Abstract.** Light is an important environmental factor controlling biogenic volatile organic compound (BVOC) emissions, but in natural conditions its impact is hard to separate from other influential factors such as temperature. We studied the light response of foliar BVOC emissions, photosynthesis and stomatal conductance on three common European tree species, namely English oak (*Quercus robur*), European beech (*Fagus sylvatica*) and two provenances of Norway spruce (*Picea abies*) in Taastrup, Denmark. Leaf scale measurements were performed on the lowest positioned branches of the tree in July 2015. Light
intensity was increased in four steps (0, 500, 1000 and 1500 $\mu$mol m$^{-2}$ s$^{-1}$), whilst other chamber conditions such as temperature, humidity and $CO_2$ levels were fixed.

Whereas the emission rate differed between individuals of the same species, the relative contributions of compounds to the total isoprenoid emission remained similar. Whilst some compounds were species specific, the compounds $\alpha$-pinene, camphene, 3-carene, limonene and eucalyptol were emitted by all of the measured tree species. Some compounds, like isoprene
and sabinene, showed an increasing emission response with increasing light intensity, whereas other compounds, like camphene, had no significant emission response to light for most of the measured trees. English oak and European beech showed high light-dependent emission fractions from isoprene and sabinene, but other emitted compounds were light-independent. For the two provenances of Norway spruce, the compounds $\alpha$-pinene, 3-carene and eucalyptol showed high light-dependent fractions for many of the measured trees. This study highlights differences between compound emissions in their
response to a change in light and a possible light independence for certain compounds, which might be valid for a wider range of tree species. This information could be of importance when improving emission models and to further emphasize the discussion regarding light or temperature dependencies for individual compounds across species.





## 1 Introduction

Biogenic volatile organic compounds (BVOCs) are produced in both marine and terrestrial environments, playing important roles in both plant survival and in the reactive chemistry of the atmosphere (Guenther et al., 1995; Goldstein and Galbally, 2007). Isoprenoids, such as isoprene (a $C_5$ unit), monoterpenes (MTs, consisting of two $C_5$ units) and sesquiterpenes (SQTs,

consisting of three $C_5$ units) contribute with approximately 68% of the total global BVOC emissions (Guenther et al., 2012). They are some of the most important BVOC groups due to their high volatility and involvement in several atmospheric reactions (Atkinson and Arey, 2003; Goldstein and Galbally, 2007, Guenther et al., 2012). The degradation of BVOCs in the air influences atmospheric processes such as production and destruction of ozone (Atkinson, 2000; Peñuelas and Staudt, 2010), but it also influences the growth of secondary organic aerosols (SOA) (Claeys et al., 2004; Ehn et al., 2014). SOA particles

are known to scatter incoming solar radiation and to act as cloud condensation nuclei, which in turn have an effect on the incoming and outgoing radiation (Laothawornkitkul et al., 2009 and references therein; Paasonen et al., 2013). In general, SOA yields are expected to be higher for compounds with internal double bonds, such as α-pinene, 3-carene, limonene and terpinolenes. However, some acyclic compounds, such as myrcene, have also been observed to produce high SOA yields (Lee et al., 2006 and references therein).

The production and release of BVOCs are sensitive to physical constraints such as light and temperature (Staudt and Bertin, 1998; Niinemets et al., 2004; Dudareva et al., 2006). Temperature controls the synthesis of isoprenoids and the diffusion rate of compounds (Niinemets et al., 2004 and references therein). The light availability determines the amount of isoprenoid precursors produced by photosynthesis and the available amount of ATP and NADPH, which are used in the $CO_2$ fixation and assimilation reactions that provide new isoprenoids (Niinemets et al., 2004 and references therein; Lichtenthaler, 2007).

However, the emission rates can also be affected by physiochemical constraints, such as stomatal conductance ($G_S$). $G_S$ can control VOC emissions temporarily in a non-steady state, when the intercellular volatile partial pressure is different from the equilibrium pressure (Niinemets and Reichstein, 2003). In a steady state, isoprene and MTs are insensitive to stomatal closure because of their high gas-phase to liquid-phase partitioning. Compounds with a large Henry's law constant (H), such as isoprene and MTs, partition to the gas phase, whilst low H compounds partition to the aqueous phase. When $G_S$ decreases, it

elevates the gas-phase partial pressure inside the stomata and increases the gradient between the intercellular air-space and atmosphere. This allows the diffusion flux of compounds with a high H to be maintained independently of stomatal conductance (Niinemets and Reichstein, 2003; Niinemets et al., 2004).

Isoprene is released upon production and therefore shows a strong direct temperature and light dependency (Kesselmeier and Staudt, 1999; Niinemets et al., 2004). The light dependency of MT emissions has, however, been more debated. In earlier

studies regarding MT emissions, a lack of light response led to the assumption that MTs were only temperature dependent (Tingey et al., 1980). Emissions of MTs were assumed to originate from internal storage structures in plants, such as resin ducts, oil glands or glandular hairs and trichomes (Fuentes et al., 1996; Kesselmeier and Staudt, 1999). The evaporation from these structures is controlled by the vapour pressure of the MTs, which in turn is affected by the air temperature and





concentration of MTs within these structures (Lerdau et al., 1997; Ghirardo et al., 2010; Taipale et al., 2011). However, more recent studies have suggested that both *de novo* and storage pool emissions can occur simultaneously. Amongst MT emitting broadleaved trees, such as Holm oak (*Quercus ilex*) and European beech (*Fagus sylvatica*), it was recognised that MT emissions were predominantly controlled by light-dependent mechanisms (Staudt and Seifert, 1995; Tollsten and Müller, 1996;

Dindorf et al., 2006). Later on, coniferous trees were also recognized to potentially emit part of their total emission as *de novo* emissions (Shao et al., 2001; Tarvainen et al., 2005; Moukhtar et al., 2006; Ghirardo et al., 2010; Taipale et al., 2011). Shao et al. (2001) measured the BVOC emissions from Scots pine (*Pinus sylvestris*) in darkness and in different light conditions. They found that MT emissions were partly influenced by photosynthetically active radiation (PAR), indicating that observed emissions originate both from storage pools and from direct biosynthesis. Ghirardo et al. (2010) used stable isotope

labelling on Norway spruce (*Picea abies*) and Scots pine and observed that the approximate contribution of *de novo* MT emissions could range between 25 and 45% for spruce and 40 and 70% for pine. Since it has been shown that light-dependent and light-independent emissions happen simultaneously, it has been suggested that the observed MT emission patterns should be regarded as a combination of light-dependent and light-independent emissions instead of only being light-independent for some species (Ghirardo et al., 2010; Taipale et al., 2011; Laffineur et al., 2011; Staudt and Lhoutellier, 2011; Song et al.,

2014).

Many emission models face the difficulty of generalizing a species or class of species into one emission potential, despite of different growing conditions and emission variabilities within species. Even though the BVOC emission patterns tend to be more similar for plants of the same species or genus, variations in emission rates have been observed. Staudt et al. (2001) screened 146 individual holm oak trees, which could be distinguished into three main types with an almost stable BVOC

composition. Their results suggest that the observed emission composition is more related to genotypic differences than to environmental impacts. Bäck et al. (2012) sampled branches from 40 mature Scots pine trees from adjacent pine stands. They could divide the trees into three chemotypes which remained fairly stable with the progression of the season. The importance of genetic diversity on observed emission patterns has been further emphasized by Persson et al. (2016) who investigated the emission patterns in genetically identical trees of English oak (*Quercus robur*), European beech and Norway spruce. Persson

et al. (2016) found differences in compound composition between two provenances of spruce, but little emission pattern differences for the remaining trees of identical genotypes.

In this study, we investigated the response of BVOC emission, photosynthetic rates and stomatal conductance of English oak, European beech and Norway spruce to different light levels. These species were chosen as they are some of the most common tree species growing in large areas within Europe (Skjøth et al., 2008) and have reported BVOC emission levels exceeding >1

30 µg gdw$^{-1}$ h$^{-1}$ (Kesselmeier et al., 1999; Dindorf et al., 2006; Holzke et al., 2006; Pokorska et al., 2012). The study aims to: (i) analyse how emissions of different BVOCs respond to changing light levels, to identify light-dependent fractions for each compound; and (ii) investigate if there are similar patterns between observed BVOC emission, photosynthetic rates and stomatal conductance. This information could be useful for our understanding of how the emission patterns of common European tree species react to changing light, which could possibly improve the algorithms used in emission models.



## 2 Methods

### 2.1 Site description and plant material

Measurements were carried out 10-31 July in 2015 at the International Phenological Garden (IPG) site Taastrup, Denmark (55°40'N, 14°30'E), maintained by the Faculty of Science at the University of Copenhagen. The IPG network performs long-term phenological observations at several sites throughout Europe on some of the most common European plant species. Each site was initially provided with up to two individuals per species. The plants used in the network are genetically identical clones, which means the genetic variation between individuals and sites is absent (Chmielewski et al., 2013). At the IPG network site at Taastrup, there are 21 trees from 13 different species and provenances with one or two individuals per species. All trees presented here were planted in 1971. Measurements were performed on two English oaks, one European beech and four Norway spruces, the latter divided into two provenances according to the framework of IPG. These provenances differ in their budburst patterns; one provenance has a budburst approximately one week earlier than the other. These provenances of spruce will henceforth be referred to as early spruce and late spruce.

During the measurement period, the weather was quite cold and humid, with an average daily temperature ranging between 13.1 and 18.8 °C and with a total rainfall of 43.6 mm during the three weeks of measurements. The average temperature and total rainfall for July 2015 was 16.4 °C and 75 mm whilst the ten-year (2006-2015) average temperature and rainfall in the area was approximately 18.2 °C and 71.8 mm (dmi.dk).

### 2.2 BVOC measurements at different light levels

Between 13 and 21 samples were taken from each tree. All measurements were made on the lowest positioned branches (1-2 m above ground) and on the southwest or south facing side of the tree using a portable photosynthesis system (Li-6400 XT, LICOR, NE, USA) equipped either with a LED source leaf chamber (6400-02B) for deciduous trees or a lighted conifer chamber (6400-22L) for the coniferous trees. The ingoing air stream (700 ml min$^{-1}$) into the chambers passed through a hydrocarbon trap and $O_3$ filter to remove organic contaminants and ozone in order to avoid BVOC oxidation before sampling. Measurements were performed during daytime (8:00-16:00). The calculations of net assimilation rates ($A_n$) and $G_S$ were performed by the instrument software, using the equations presented by von Caemmerer and Farquhar (1981). All measurements were made under fixed environmental conditions. Each leaf or needle twig was acclimated to 400 µmol $CO_2$ mol$^{-1}$ air and 50-60% relative humidity for one hour before BVOC emission sampling. The temperature within the chamber was set according to the anticipated average daily temperature (18-23 °C during the campaign) in order to minimize potential stress emissions from the plant. Each leaf or needle twig was measured under four light levels (0, 500, 1000 and 1500 µmol m$^{-2}$ s$^{-1}$) by stepwise increasing PAR from 0 µmol m$^{-2}$ s$^{-1}$ to 1500 µmol m$^{-2}$ s$^{-1}$. This direction was chosen in order to mimic the daily increase in light intensity. After the first acclimation period of one hour at 0 µmol m$^{-2}$ s$^{-1}$, an additional 30-minute acclimation period was applied after switching to a new light level in order to ensure that the leaf or needle twig had adjusted to the new conditions. This acclimation time was chosen based on preliminary tests showing that leaf photosynthesis remained



reasonably stable after 30 minutes' adjustment to the new light intensity. The BVOC emissions from the trees were collected by extracting air from the chamber outlets into stainless steel cartridges (Markes International Limited, Llantrisant, UK) packed with adsorbents Tenax TA (a porous organic polymer) and Carbograph 1TD (graphitized carbon black). The air extraction was performed using flow-controlled pocket pumps (SKC Ltd., Dorset, UK) with a flow rate of 200 ml min$^{-1}$. Empty chamber

blanks were collected every second day with the same chamber conditions in order to account for possible background contamination in the measured samples.

### 2.3 BVOC analysis

The BVOC sample cartridges were sealed with Teflon coated brass caps directly after sampling, stored at 3 °C and analysed within eight weeks. A gas chromatograph-mass spectrometer (7890A Series GC coupled with a 5975C inert MSD/DS

Performance Turbo EI system, Agilent, Santa Clara, CA, USA) was used for analysis after thermal desorption (UNITY2 coupled with an ULTRA autosampler, Markes, Llantrisant, UK). The oven temperature was held at 40 °C for 1 min, raised to 210 °C in steps of 5 °C min$^{-1}$ and lastly up to 250 °C in steps of 20 °C min$^{-1}$. Helium was used as the carrier gas and the BVOC separation was done with a HP-5 capillary column (50 m, diameter 0.2 mm and film thickness 0.33 µm). The identification and quantification of BVOCs was done using pure standard solutions for isoprene, α-pinene, camphene, β-pinene, δ-

phellandrene, ρ-cymene, 1,8-cineole, ocimene, γ-terpinene, terpinolene, linalool, aromadendrene, α-humulene and nerolidol in methanol (Fluka, Buchs, Switzerland). These standard solutions were injected into adsorbent cartridges in a stream of helium. If there was a compound detected without an available standard, it was identified according to the mass spectra in the NIST library, and quantified using α-pinene for MTs and α-humulene for SQTs. The sample chromatograms were analysed with the MSD Chemstation Data Analysis software (G1701CA C.00.00 21 Dec 1999; Agilent Technologies, Santa Clara, CA,

USA). Compounds that were found in the empty chamber blanks collected in the field were subtracted from the samples. Only isoprenoids were analysed in this study. Emissions were calculated by using the emission rate equation for the dynamic enclosure technique presented by Ortega and Helmig (2008). For each of the three light levels above 0, the light-dependent fraction of the total compound emission was calculated [as 100% × (light emission - dark emission)/light emission] and used as an indicator for its emission response to changing light. The values ranged from 0% (no light-dependence) to 100%

(compound emitted entirely light-dependently).

### 2.4 Statistical analysis

Repeated measures ANOVA tests were computed in the Rstudio software (Rstudio team, 2015, version 0.99.491) in order to test if the observed emission rates of each compound and the $A_n$ or $G_S$ rates differed statistically between the light levels. If a significant effect of light was observed, a simple a priori contrast was used to test which light level was significantly different

from the dark measurements. The statistical analyses were done separately for each tree species.





## 3 Results

### 3.1 BVOC emission from English oak

Figure 1 shows the total BVOC emission rate and the compound contributions of the two English oaks at different light levels. The first oak had a statistically significant increase of the total emission across light levels, whilst the emission rate of the second oak saturated at 1000 µmol m$^{-2}$ s$^{-1}$ (Table 1). Between one and seven compounds were detected at the measured light levels and the detected compounds were isoprene, tricyclene, α-pinene, camphene, 3-carene, limonene and eucalyptol. The main emitted compound was isoprene, with no emission during darkness and an emission rate between 2.3-19.8 µg gdw$^{-1}$ h$^{-1}$ for oak 1 and 1.3-9.3 µg gdw$^{-1}$ h$^{-1}$ for oak 2 at light levels of 500-1500 µmol m$^{-2}$ s$^{-1}$. The relative contribution of isoprene to the total emission with light levels at or above 500 µmol m$^{-2}$ s$^{-1}$ was >96% (Fig. 1). At a light level of 0 µmol m$^{-2}$ s$^{-1}$, the main detected compounds were limonene and α-pinene. The emissions of these MTs remained stable across measured PAR levels, with emission rates of <0.1 µg gdw$^{-1}$ h$^{-1}$ at all levels (see Appendix A for absolute values).

### 3.2 BVOC emission from European beech

In contrast to English oak, European beech showed a smaller and non-significant response of the total isoprenoid emission rate to a change in light (Table 1, Fig. 2). Beech emitted between one and five detected isoprenoids in darkness and between four and eight with light. Detected compounds were tricyclene, α-pinene, camphene, sabinene, 3-carene, limonene, eucalyptol and caryophyllene. Sabinene was not detected at 0 µmol m$^{-2}$ s$^{-1}$ but was the main emitted compound with light, increasing from 66% of the total emission at 500 µmol m$^{-2}$ s$^{-1}$ to 76% at 1500 µmol m$^{-2}$ s$^{-1}$. Limonene was the main emitted compound in darkness. The amount of limonene released remained fairly stable across the studied light levels and ranged between 0.06 and 0.09 µg gdw$^{-1}$ h$^{-1}$. The other emitted MTs did not change their emission patterns with increasing light. At light levels 1000 and 1500 µmol m$^{-2}$ s$^{-1}$, the SQT caryophyllene was released, with the highest emissions at 1500 µg gdw$^{-1}$ h$^{-1}$ (see Appendix A for absolute values, Fig. 2).

### 3.3 BVOC emission from Norway spruce

Figures 3a and b show the emission rate and the compound contribution with increasing light levels for early spruce and late spruce, respectively. All four spruce trees emitted isoprene with light (P<0.001 for early spruce 1 and late spruce 1 and 2, P>0.1 for early spruce 2) with a contribution to the total emission of 30-65%. In contrast, limonene and α-pinene were emitted both in darkness as well as with light, but with lower absolute emissions in darkness (see Appendix A for absolute values, Fig. 3). For early spruce, between four and nine isoprenoids were detected. These were isoprene, tricyclene, α-pinene, camphene, β-pinene, 3-carene, limonene, eucalyptol, linalool, α-farnesene and β-farnesene. Only one of the two early spruce trees emitted linalool and SQTs. The main detected compound for both trees was isoprene, followed by limonene. The total emission from early spruce 1 saturated at 500 µmol m$^{-2}$ s$^{-1}$ with no significant change with increasing light (P>0.1), whilst early spruce 2 decreased its total emission at 1000 µmol m$^{-2}$ s$^{-1}$ and then increased again somewhat at 1500 µmol m$^{-2}$ s$^{-1}$ (Fig. 3a). Late spruce



emitted two to ten isoprenoids at all light levels and the detected compounds were isoprene, tricyclene, α-pinene, camphene, β-pinene, 3-carene, α-terpinene, limonene, eucalyptol and γ-terpinene. β-pinene was emitted by both provenances of Norway spruce, but with higher emissions rates from late spruce in combination with higher emissions of α-pinene. Only late spruce 1 emitted tricyclene and α-terpinene and only at PAR levels of 1000 and 1500 µmol m$^{-2}$ s$^{-1}$. Both trees had an increase in total emission up to 1000 µmol m$^{-2}$ s$^{-1}$, with a decrease in emissions at 1500 µmol m$^{-2}$ s$^{-1}$ for late spruce 1. Late spruce 1 reached its peak emission of 2.2 µg gdw$^{-1}$ h$^{-1}$ at 1000 µmol m$^{-2}$ s$^{-1}$, whilst late spruce 2 had a stable emission between 0.6-0.9 µg gdw$^{-1}$ h$^{-1}$ with light. The emitted compounds from late spruce 1 followed a similar emission pattern as the total emission rate, but for late spruce 2 all compounds except α-pinene, eucalyptol and γ-terpinene remained fairly stable with increase in light (Appendix A, Fig. 3b).

## 3.4 Light-dependent fractions of different compounds

Whilst some compounds like isoprene and sabinene were specific for different tree species, the compounds α-pinene, camphene, 3-carene, limonene and eucalyptol were emitted from all of the measured leaves or needle twigs. As these compounds were emitted at different light levels, we will assess the light dependency of these compounds. The light-dependent fraction for isoprene was 100% for all of the isoprene emitting trees (Table 2). The same fraction and significance were also found for sabinene emission from beech ($P<0.001$, Table 2). The light-dependent fraction of other MTs however depended on the compound and the tree species. Camphene had a significant change in emission from darkness to 500 µmol m$^{-2}$ s$^{-1}$ for early and late spruce 2, but for remaining light levels camphene showed no clear light dependency for any of the measured trees (Appendix A, Table 1 and 2).

For the oaks, besides isoprene no other compounds showed a significant light dependency. For beech, some compounds like camphene, 3-carene, limonene and eucalyptol increased the light-dependent fractions with higher light levels, but without this being a significant increase in its emissions (Appendix A, Table 1 and 2).

The two provenances of spruce showed a higher light-dependent fraction for MTs in comparison to the broadleaved trees. Early spruce 1 and late spruce 1 showed light-dependent fractions of 76-86% and 67-94% respectively for the total MT emission (Table 2). Both trees had high light-dependent fractions for the compounds α-pinene, 3-carene and eucalyptol. For early spruce 1, eucalyptol increased its light-dependent fraction with increasing light levels. For late spruce 1 there was a higher percentage of light dependency for α-pinene, but only limonene increased in light dependency with increasing light. Early spruce 2 had low light-dependent fractions for all compounds except eucalyptol, whilst late spruce 2 had high light-dependent fractions for α-pinene and eucalyptol. Although several of the above mentioned compounds from early spruce 2 and late spruce 2 showed a light dependency, this light dependency did not change with a change in light level (Table 2).



### 3.5 Photosynthesis and stomatal conductance of oak, beech and spruce

For oak, the assimilation ($A_n$) rates were fairly similar between the two trees, ranging from -0.6 - -0.5 µmol $CO_2$ $m^{-2}$ $s^{-1}$ in darkness and 2.4-4.5 µmol $CO_2$ $m^{-2}$ $s^{-1}$ with light (Fig. 4a, Table 1). The difference was larger for the stomatal conductance ($G_S$): oak 1 showed a significant difference with increasing light ($P<0.05$) in comparison to oak 2 which showed higher internal

variation ($P>0.2$). For beech, $A_n$ increased from darkness to the PAR level of 500 µmol $m^{-2}$ $s^{-1}$ ($P<0.001$), but did not show a response to further increase in light ($P>0.6$). $A_n$ was between 3-3.6 µmol $CO_2$ $m^{-2}$ $s^{-1}$ with light and -0.3 µmol $CO_2$ $m^{-2}$ $s^{-1}$ in darkness, whilst $G_S$ ranged between 100-400 mmol $H_2O$ $m^{-2}$ $s^{-1}$ for all light levels (Fig. 4b).

For early spruce 1, $A_n$ was between 9.5-11.3 µmol $CO_2$ $m^{-2}$ $s^{-1}$ at a light level of 500 and 1000 µmol $m^{-2}$ $s^{-1}$ which decreased to 7.3 µmol $CO_2$ $m^{-2}$ $s^{-1}$ at 1500 µmol $m^{-2}$ $s^{-1}$. $G_S$ followed a similar pattern, ranging from 1000-1200 mmol $H_2O$ $m^{-2}$ $s^{-1}$ at a

light level of 500 and 1000 µmol $m^{-2}$ $s^{-1}$ and decreased to 700 mmol $H_2O$ $m^{-2}$ $s^{-1}$ at a light level of 1500 µmol $m^{-2}$ $s^{-1}$ (Fig. 4c, Table 1). A similar pattern as the BVOC emissions for early spruce 2 could also be seen in the rates of $A_n$ and $G_S$ with lower values coinciding with lower emissions, but which was significant only for $A_n$ (Table 1). Late spruce 1 had a higher emission rate in comparison to late spruce 2, which was also evident for the $A_n$ and $G_S$ rates. Whilst late spruce 1 showed an increase in both $A_n$ and $G_S$ with increasing light levels ($P<0.05$), late spruce 2 did not show any clear response to increasing light above

500 µmol $m^{-2}$ $s^{-1}$ ($P>0.2$). Late spruce 1 had an average $A_n$ rate of 4.5-10.9 µmol $CO_2$ $m^{-2}$ $s^{-1}$ and an average $G_S$ rate of 400-1100 mmol $H_2O$ $m^{-2}$ $s^{-1}$ with light. For the second spruce, the $A_n$ and $G_S$ rates were stable at an average range of 3.6-5.1 µmol $CO_2$ $m^{-2}$ $s^{-1}$ and 300-500 mmol $H_2O$ $m^{-2}$ $s^{-1}$ with light (Fig. 4d).

Overall, the investigated trees showed a similar response to light in their light-dependent BVOC emissions, $A_n$ and $G_S$, but the light level at which these processes saturate could vary for individual leaves or needle twigs.

### 4 Discussion

Light plays an important role as a driver of BVOC emissions, particularly in regards to *de novo* emissions. However, few studies have investigated in situ if the compounds emitted from different tree species respond similarly with a conditional change in light. Our aim was to investigate how different compounds responded to changing light conditions and if the response was similar between different tree species.

### 4.1 Responses of BVOC emissions to changing light conditions

Some compounds were species specific in regard to their emission rates, with high emissions of isoprene from English oak, sabinene from European beech and either α-pinene or limonene by the provenances of Norway spruce. However, the compounds α-pinene, camphene, 3-carene, limonene and eucalyptol were emitted by all species, which made it possible to study how the light dependency of these compounds would differ between selected tree species.

All the trees with the capacity to emit isoprene showed a clear response to light, which has been confirmed by other studies as well (Tingey et al., 1981; Lehning et al., 1999; Grabmer et al., 2006). However, the light dependency of other compounds than



isoprene differed depending on the tree species. The compound camphene showed significant emission responses from early spruce 2 and late spruce 2, but only going from darkness to 500 µmol m$^{-2}$ s$^{-1}$. For the remaining trees, there was no clear camphene emission response to an increase in light. This suggests that this compound should be considered to be light-independent when emission rates are to be modelled. Emission of camphene has been shown to be temperature dependent in

a study on *Abies alba* (Moukhtar et al., 2006).

A similarity which was found for the oaks and the beech was that apart from their main emitted compounds, the emission rate of other MTs did not show any significant response with increasing light (P>0.05). This observation would suggest that the emission of MTs from these deciduous trees should be regarded as light-independent instead of light-dependent, dividing the emissions into light-dependent and light-independent fractions. Dividing the emitted compounds into light dependency

fractions has also been suggested for Norway spruce, for which the light-dependent emissions have been reported to range between 25-45% of the total emission rates (Ghirardo et al., 2010).

The two provenances of spruce had different responses of their emitted compounds with an increase in light. Early spruce 1 showed light-dependent fractions from α-pinene, 3-carene, limonene and eucalyptol, but with eucalyptol being the only MT compound which continued to increase its light-dependent fraction with increasing light intensity. A similar light dependency

of eucalyptol has also been found for emissions from *Abies alba* (Moukhtar et al., 2006). Early spruce 2 showed light-dependent fractions from α-pinene, camphene and eucalyptol. However, as the amount of samples taken on early spruce 2 were few, it is difficult to draw any clear conclusions for this tree. Both the late spruce trees had light-dependent emissions of α-pinene and eucalyptol. Late spruce 1 also showed light-dependent fractions for 3-carene going from darkness to light, but the overall emission rate of this compound was low and of little importance in regard to the general compound contribution.

For late spruce 2, α-pinene and camphene showed significant emission increases from darkness to 500 µmol m$^{-2}$ s$^{-1}$. The response of late spruce 2 might however be masked by high internal emission variation at 500 µmol m$^{-2}$ s$^{-1}$.

With the current experimental setup, it is only possible to make assumptions of the relative contributions of *de novo* sources and storage pools. This is otherwise often tested by using $^{13}CO_2$ labelling, where *de novo* emissions would have $^{13}C$ incorporated into their compound structures after a pulse of labelled $^{13}CO_2$ (Ghirardo et al., 2010). But by using genetically

identical trees and fixed environmental conditions inside the measurement chamber, it has been possible to study the emission response of different compounds to an increase in light intensity. Our study shows that different compounds respond differently to a change in light and that compounds like camphene have similar emission responses for English oak, European beech and Norway spruce and that all of the measured trees released isoprenoids in darkness, with emissions ranging from 0-0.4 µg gdw$^{-1}$ h$^{-1}$ for the broadleaf trees and 0.01-0.22 µg gdw$^{-1}$ h$^{-1}$ for the provenances of spruce. This would indicate that species such as

English oak and European beech, which are considered to lack specific storage compartments, have a capacity to store compounds in the mesophyll, which has also been suggested by other studies (Niinemets and Reichstein, 2003; Holopainen and Gershenzon, 2011). In a study by Loreto et al. (2000), $^{13}C$ labelling was used on Holm oak (*Quercus ilex*) with and without illumination and found that the newly synthesized compounds could continue to be emitted long after initiation of darkness. It was suggested that the volatile compounds could be non-specifically stored within the plant leaves, either in the lipid phase or





in the aqueous phase. Furthermore, Bäck et al. (2005) did a modelling study on Scots pine where a mesophyll pool was included, which enabled them to better capture diurnal and seasonal emission trends of MT emissions. These results suggest that as there might exist non-specific storage within the leaf tissue, *de novo* emitting tree species need to be considered to have storage pools in emission models as well.

As many models divide plants into categories or plant functional types depending on the growing conditions they have adapted to (Schurgers et al., 2011; Guenther et al., 2012), an approach looking at the emission patterns of separate compounds would perhaps improve the division of plant functional types further. If the plants are also categorized into the compound emission response, the model would perhaps provide more realistic values by dividing the compounds into light-dependent or independent fractions. The possibility of a converging behaviour in light dependences between different species is a promising idea, but further investigations are necessary to confirm this suggestion. We would therefore strongly suggest that more studies assessing light dependency of different compounds are performed on similar or different tree species in order to verify this light dependency of the compounds.

### 4.2 Emission pattern variation and shade adaptation of the leaves and needle twigs

The European tree species presented here have distinct emission patterns: English oak is a known high isoprene emitter, European beech mainly emits MTs such as sabinene, and Norway spruce is known to emit both isoprene and MTs (Dindorf et al., 2006; Ghirardo et al., 2010; Pokorska et al., 2012). The English oak clones in this study had emission rates between 3.5-18.3 µg gdw$^{-1}$ h$^{-1}$ at a light level of 1000 µmol m$^{-2}$ s$^{-1}$ and a set temperature range of 18-21 °C. These emission rates are in line with the standardized emission rates reported by previous studies (Isidorov et al., 1985; Kesselmeier and Staudt, 1999; Pokorska et al., 2012; Persson et al., 2016). In regards to their photosynthetic and stomatal conductance ranges, they are comparable with studies performed on oak leaves grown in either shaded or semi-shaded conditions (Morecroft and Roberts, 1999; Valladares et al., 2002). Between 96-99% of the total emission for oak consisted of isoprene, followed by MTs such as limonene and α-pinene. This compound contribution has not only been stable over three years of measurements on these genetically identical trees, but it is also in agreement with measurements at other sites (Staudt et al., 2001; Persson et al., 2016; van Meeningen et al., 2016). The low emission pattern variation would suggest that even if environmental factors such as temperature or light influence the total emission from oak, these do not alter the compound contribution to a great extent (Staudt et al., 2001; van Meeningen et al., 2016).

The European beech had an average total emission of 0.3 µg gdw$^{-1}$ h$^{-1}$ in darkness and between 0.8-1.0 µg gdw$^{-1}$ h$^{-1}$ with light. However, there were big differences in emission amounts between leaves, making it difficult to see any clear increase in BVOC emissions with an increase in light. When the light level exceeded 1000 µmol m$^{-2}$ s$^{-1}$, there was also an increase in SQT emissions. But as there was no obvious sign of injury and due to the low contribution of detected SQTs (<10%), we consider the leaves to be unstressed. The emission rates are in the lower ranges in comparison to other studies with standardized emission rates (Moukhtar et al., 2005; Dindorf et al., 2006 and references therein). This could be because all samples were taken on the lowest positioned branches of the tree. In the study from the same site performed in 2013, the emission rates were taken at





three different height levels within the canopy of all the above mentioned trees. For the European beech, the emission rates were much higher at the top of the canopy in comparison to lower levels, with an average standardized emission of 26.5 µg gdw$^{-1}$ h$^{-1}$ at the top of the canopy and 3.6 µg gdw$^{-1}$ h$^{-1}$ at the bottom. The lower emission rate found in this study could be caused by more shade-adapted leaves, with a possible lower capacity to respond to high increases in light. The levels of $A_n$

and $G_S$ presented here are comparable with other studies performed on leaves adapted to shaded or semi-shaded conditions (Valladares et al., 2002; Warren et al., 2007; Scartazza et al., 2016). As the emission increase was unclear for the chosen light levels, more light levels between 0-500 µmol m$^{-2}$ s$^{-1}$ would be preferable. It would also be advisable to make more measurements at the top of the canopy in comparison to the lower levels in order to not underestimate the emission potentials for European beech.

There were distinct differences in emission spectra between the two provenances of Norway spruce. The main emitted compound for both provenances was isoprene, but regarding the emitted MTs early spruce was mainly a limonene emitter whilst late spruce emitted α-pinene. This emission pattern difference remained stable over three summer seasons for this site (Persson et al., 2016 for 2013, unpublished data for 2014, current study for 2015). Furthermore, late spruce also emitted β-pinene at a higher rate than the early spruce trees, whilst the compounds α-terpinene and γ-terpinene were only emitted by late

spruce. This would suggest that for different provenances of the same species, different compound adaptations might exist. Studies on other tree species have suggested that trees can be divided into chemotypes depending on their emission patterns and that the compound contribution of these chemotypes remains fairly stable over time (Staudt et al., 2001; Bäck et al., 2012). This result is not surprising as trees would need to be able to adapt according to the local growing conditions.

The average emission rates at 1000 µmol m$^{-2}$ s$^{-1}$ ranged between 0.1-0.6 µg gdw$^{-1}$ h$^{-1}$ for early spruce and 0.9-2.2 µg gdw$^{-1}$ h$^{-1}$

for late spruce, which were in range of previous studies (Kesselmeier and Staudt, 1999; Grabmer et al., 2006). The four light levels that were tested did not provide enough information to address the light response entirely. More points taken between 0-500 µmol m$^{-2}$ s$^{-1}$ would therefore be advisable in order to fully understand the change in emission amounts. The second early spruce tree showed more fluctuation between different light levels, possibly as a response to stress exposure. When measurements were performed on this tree in 2013, the needles on the lowest branches dried and fell off after a prolonged

period without rain in the middle of July (Persson et al., 2016). In 2014, when measurements were performed again, the lower twigs had still not recovered and it was not possible to make any measurements on that level (unpublished data). In 2015, new twigs had started to emerge again on early spruce 2, but twigs were small and visibly less healthy. With less material to make measurements on and with possible recovery from stress, it is difficult to fully capture the release of BVOC emission from early spruce 2. The average $A_n$ rates for early spruce and late spruce were between 4.3-12.1 and 3.6-12 µmol $CO_2$ m$^{-2}$ s$^{-1}$

respectively, whilst the $G_S$ rates ranged between 400-1200 mmol $H_2O$ m$^{-2}$ s$^{-1}$ for early spruce and between 300-1000 mmol $H_2O$ m$^{-2}$ s$^{-1}$ for late spruce. These values are in range or slightly higher than reported in other studies (Le Thiec et al., 1994; Roberntz and Stockfors, 1998; Špunda et al., 2005). Early spruce 1 and late spruce 2 behaved in a similar fashion as European beech with a tendency to stabilize their $A_n$ and $G_S$ rates at a light level of 500 µmol m$^{-2}$ s$^{-1}$, indicating some shade adaptation of the selected needle twigs. Late spruce 1 increased both in $A_n$ and $G_S$ rates with light, possible because the tree stands more





exposed than the others in the northeast corner of the IPG site and therefore is more light adapted in comparison to the other trees. Early spruce 2 showed the same fluctuating pattern in $A_n$ and $G_S$ rates as with the observed BVOC emissions, most likely due to a restricted sample size and previous effect of drought stress on the tree.

## 5 Summary and conclusions

Measurements were performed on one European beech and on genetically identical mature individuals of English oak and two provenances of Norway spruce with the aim to study the light response of the emitted compounds. Our study shows that, despite the existence of differences in emission amounts, the relative contribution of the main emitted compounds was similar between the individuals of the same tree species. Compounds like isoprene showed a light dependency for all of the measured isoprene-emitting trees, whilst camphene showed a slight response from early spruce 2 and late spruce 2 going from darkness

to 500 µmol m$^{-2}$ s$^{-1}$ but no significant response for the remaining trees. Apart from isoprene for English oak and sabinene for European beech, there was no clear light dependency of other emitted isoprenoids which could show a possible convergence in the response of these minor compounds to changes in light. For the provenances of spruce, some compounds like α-pinene, 3-carene and eucalyptol showed high light-dependent fractions for many of the individuals, which remained fairly stable with increasing light. This would possibly suggest that some MT compounds should be considered to be light-dependent in regard

to emission models. As all measurements were performed on the lowest positioned branches of the tree, some trees showed indications of shade adaptation which could perhaps have inhibited the light response of certain compounds. The low sample size could also be responsible for the difficulty in finding statistically significant increases of emissions with light. However, the study does show a potential convergence of the light responses for compounds such as camphene for all the studied trees and monoterpene emission from English oak and European beech. This convergence needs to be studied further both for the

mentioned compounds and for other tree species in order to fill in potential knowledge gaps, but we believe that this could possibly be of significance to improve emission modelling.



## Appendix

**Table A1. The mean average actual emission (± standard deviation) of detected compounds at light levels (PAR) 0, 500, 1000 and 1500 µmol m$^{-2}$ s$^{-1}$ and the number of samples taken from English oak (*Quercus robur*), European beech (*Fagus sylvatica*) and the two provenances of spruce (*Picea abies*) in µg gdw$^{-1}$ h$^{-1}$. No data (n.d.) indicates that the compound was not detected in any sample for the measured light level on that particular tree.**

| Tree | PAR | Isoprene | Tricyclene | α-pinene | Camphene | Sabinene | β-pinene | 3-Carene | α-terpinene | Limonene | Eucalyptol | γ-terpinene | Linalool | SQT | Total |
|---|---|---|---|---|---|---|---|---|---|---|---|---|---|---|---|
| Oak 1 (n=15) | 0 | n.d. | <0.01 | 0.02 ± <0.01 | 0.03 ± 0.02 | n.d. | n.d. | 0.02 ± <0.01 | n.d. | 0.08 ± 0.04 | 0.02 ± <0.01 | n.d. | n.d. | n.d. | 0.16 ± 0.08 |
| | 500 | 5.08 ± 2.32 | <0.01 ± <0.01 | 0.01 ± 0.01 | 0.01 ± 0.01 | n.d. | n.d. | 0.01 ± <0.01 | n.d. | 0.05 ± 0.03 | 0.01 ± 0.01 | n.d. | n.d. | n.d. | 5.19 ± 2.27 |
| | 1000 | 12.53 ± 3.68 | 0.01 ± 0.01 | 0.01 ± <0.01 | 0.01 ± 0.01 | n.d. | n.d. | 0.01 ± 0.01 | n.d. | 0.05 ± 0.03 | 0.01 ± <0.01 | n.d. | n.d. | n.d. | 12.62 ± 3.65 |
| | 1500 | 16.31 ± 2.91 | 0.01 ± <0.01 | 0.01 ± <0.01 | 0.02 ± 0.01 | n.d. | n.d. | 0.01 ± <0.01 | n.d. | 0.06 ± 0.03 | 0.01 ± <0.01 | n.d. | n.d. | n.d. | 16.43 ± 2.86 |
| Oak 2 (n=17) | 0 | 0.02 | <0.01 | 0.01 ± <0.01 | 0.01 ± <0.01 | n.d. | n.d. | 0.01 ± <0.01 | n.d. | 0.03 ± 0.01 | 0.01 ± <0.01 | n.d. | n.d. | n.d. | 0.05 ± 0.03 |
| | 500 | 2.68 ± 0.99 | <0.01 | 0.01 ± 0.01 | 0.04 ± 0.02 | n.d. | n.d. | 0.01 ± 0.01 | n.d. | 0.08 ± 0.04 | 0.02 ± 0.01 | n.d. | n.d. | n.d. | 2.79 ± 1.01 |
| | 1000 | 6.53 ± 2.0 | 0.01 | 0.02 ± 0.01 | 0.05 ± 0.02 | n.d. | n.d. | 0.01 ± <0.01 | n.d. | 0.09 ± 0.02 | 0.02 ± <0.01 | n.d. | n.d. | n.d. | 6.68 ± 1.95 |
| | 1500 | 5.68 ± 2.22 | 0.01 | 0.03 ± <0.01 | 0.04 ± 0.03 | n.d. | n.d. | 0.01 ± 0.01 | n.d. | 0.08 ± 0.01 | 0.03 ± <0.01 | n.d. | n.d. | n.d. | 5.79 ± 2.14 |
| Beech (n=21) | 0 | n.d. | n.d. | 0.04 ± 0.03 | 0.05 ± 0.03 | n.d. | n.d. | 0.04 ± 0.03 | n.d. | 0.09 ± 0.05 | 0.03 ± 0.01 | n.d. | n.d. | n.d. | 0.25 ± 0.14 |
| | 500 | n.d. | 0.01 ± 0.02 | 0.04 ± 0.02 | 0.06 ± 0.03 | 0.52 ± 0.78 | n.d. | 0.03 ± 0.02 | n.d. | 0.09 ± 0.07 | 0.04 ± 0.01 | n.d. | n.d. | n.d. | 0.79 ± 0.76 |
| | 1000 | n.d. | 0.01 | 0.06 ± 0.06 | 0.04 ± 0.02 | 0.65 ± 0.97 | n.d. | 0.03 ± 0.03 | n.d. | 0.06 ± 0.05 | 0.03 ± 0.01 | n.d. | n.d. | 0.02 | 1.23 ± 1.18 |
| | 1500 | n.d. | 0.01 | 0.03 ± 0.03 | 0.03 ± 0.02 | 0.75 ± 1.05 | n.d. | 0.03 ± 0.02 | n.d. | 0.07 ± 0.03 | 0.03 ± 0.01 | n.d. | n.d. | 0.04 ± 0.06 | 0.99 ± 1.05 |
| Early spruce 1 (n=14) | 0 | n.d. | n.d. | 0.01 ± 0.01 | 0.02 ± 0.01 | n.d. | n.d. | <0.01 ± <0.01 | n.d. | 0.01 ± 0.01 | 0.01 ± <0.01 | n.d. | n.d. | n.d. | 0.05 ± 0.03 |
| | 500 | 0.18 ± 0.03 | n.d. | 0.03 ± <0.01 | 0.02 ± 0.01 | n.d. | 0.02 ± 0.01 | 0.06 ± 0.03 | n.d. | 0.08 ± 0.01 | 0.04 ± 0.01 | n.d. | n.d. | 0.10 ± 0.07 | 0.54 ± 0.04 |
| | 1000 | 0.21 ± 0.02 | n.d. | 0.03 ± 0.01 | 0.02 ± 0.02 | n.d. | 0.01 ± 0.01 | 0.04 ± 0.02 | n.d. | 0.06 ± 0.01 | 0.05 ± 0.01 | n.d. | n.d. | 0.16 ± 0.08 | 0.58 ± 0.09 |
| | 1500 | 0.25 ± 0.03 | <0.01 | 0.03 ± <0.01 | 0.02 ± 0.01 | n.d. | 0.02 ± <0.01 | 0.01 ± <0.01 | n.d. | 0.05 ± <0.01 | 0.06 ± <0.01 | n.d. | n.d. | 0.14 ± 0.01 | 0.60 ± 0.05 |




| | | | | | | | | | | | | | | | |
|---|---|---|---|---|---|---|---|---|---|---|---|---|---|---|---|
| Early spruce 2 (n=13) | 0 | n.d. | <0.01 ± <0.01 | 0.02 ± <0.01 | 0.02 ± <0.01 | n.d. | n.d. | 0.01 ± <0.01 | n.d. | 0.04 ± 0.02 | 0.01 ± 0.01 | n.d. | n.d. | n.d. | 0.10 ± 0.04 |
| | 500 | 0.23 ± <0.01 | n.d. | 0.02 ± <0.01 | <0.01 ± <0.01 | n.d. | n.d. | 0.01 ± <0.01 | n.d. | 0.07 ± 0.01 | 0.04 ± <0.01 | n.d. | n.d. | n.d. | 0.37 ± 0.02 |
| | 1000 | 0.05 | n.d. | 0.01 ± <0.01 | 0.01 ± 0.01 | n.d. | n.d. | 0.01 ± <0.01 | n.d. | 0.01 | 0.02 ± 0.02 | n.d. | n.d. | n.d. | 0.12 ± 0.13 |
| | 1500 | 0.14 ± 0.05 | n.d. | 0.03 ± <0.01 | <0.01 ± <0.01 | n.d. | 0.02 ± <0.01 | 0.01 ± <0.01 | n.d. | 0.04 ± <0.01 | 0.05 ± <0.01 | n.d. | n.d. | n.d. | 0.29 ± 0.05 |
| Late spruce 1 (n=13) | 0 | n.d. | n.d. | <0.01 | 0.03 ± 0.02 | n.d. | n.d. | n.d. | n.d. | 0.01 ± 0.01 | <0.01 | n.d. | n.d. | n.d. | 0.04 ± 0.03 |
| | 500 | 0.31 ± 0.13 | <0.01 | 0.09 ± 0.09 | 0.02 ± 0.01 | n.d. | 0.01 | 0.01 | 0.01 | 0.02 ± 0.03 | 0.01 ± 0.01 | n.d. | n.d. | n.d. | 0.49 ± 0.30 |
| | 1000 | 1.26 ± 0.49 | 0.01 ± 0.01 | 0.29 ± 0.26 | 0.21 ± 0.27 | n.d. | 0.07 ± 0.03 | 0.02 ± 0.01 | 0.04 ± 0.02 | 0.05 ± 0.05 | 0.10 ± 0.05 | 0.10 ± 0.06 | n.d. | n.d. | 2.16 ± 0.84 |
| | 1500 | 0.54 ± 0.04 | <0.01 | 0.23 ± 0.01 | 0.01 ± <0.01 | n.d. | 0.03 ± <0.01 | 0.01 ± <0.01 | 0.02 ± <0.01 | 0.04 ± <0.01 | 0.03 ± <0.01 | 0.03 ± <0.01 | n.d. | n.d. | 0.93 ± 0.04 |
| Late spruce 2 (n=18) | 0 | n.d. | n.d. | 0.02 ± 0.01 | 0.03 ± 0.02 | n.d. | n.d. | 0.04 ± 0.01 | n.d. | 0.04 ± 0.02 | 0.01 ± 0.01 | n.d. | n.d. | n.d. | 0.25 ± 0.12 |
| | 500 | 0.30 ± 0.16 | n.d. | 0.14 ± 0.07 | <0.01 ± <0.01 | n.d. | 0.02 ± 0.02 | 0.05 ± 0.04 | n.d. | 0.06 ± 0.05 | 0.02 ± 0.02 | n.d. | n.d. | n.d. | 1.17 ± 0.70 |
| | 1000 | 0.45 ± 0.06 | n.d. | 0.20 ± 0.01 | <0.01 ± <0.01 | n.d. | 0.03 ± <0.01 | 0.06 ± <0.01 | n.d. | 0.06 ± 0.01 | 0.03 ± <0.01 | 0.02 ± 0.02 | n.d. | n.d. | 1.69 ± 0.12 |
| | 1500 | 0.33 ± 0.03 | n.d. | 0.015 ± 0.04 | <0.01 ± <0.01 | n.d. | 0.03 ± 0.01 | 0.05 ± 0.01 | n.d. | 0.05 ± 0.01 | 0.04 ± <0.01 | 0.04 ± <0.01 | n.d. | n.d. | 1.41 ± 0.08 |

## Acknowledgements

The authors would like to thank Vetenskapsrådet (VR 621-2011-3190) for partly funding the project. We are grateful to Anders K. Nørgaard and the rest of the staff at the experimental farms in Taastrup, University of Copenhagen for their support in the field and for providing facilities. We are also grateful to Gosha Sylvester at the University of Copenhagen for performing BVOC sample analysis. Magnus Kramshøj, Frida Lindwall, Jing Tang, Michelle Schollert Reneerkens and Janne Rinne provided valuable comments on the manuscript. The study was performed within the framework of LUCCI, which is a research centre at Lund University for studies of carbon cycles and climate interaction.





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





**Table 1: The P-values from repeated measures ANOVA tests on the emission rate of each compound, photosynthetic rates ($A_n$) and stomatal conductance ($G_S$) in response to an increase in light intensity. The trees that were measured were two individuals of English oak (*Quercus robur*), one European beech (*Fagus sylvatica*), two individuals of Norway spruce (*Picea abies*) with an early budburst (Early spruce) and two individuals of Norway spruce with a late budburst (Late spruce). P-values marked in bold show statistically significant values (P<0.05). Isoprene was not detected from the European beech tree.**

| Compound | Oak 1 | Oak 2 | Beech | Early spruce 1 | Early spruce 2 | Late spruce 1 | Late spruce 2 |
|---|---|---|---|---|---|---|---|
| Isoprene | **<0.001** | **<0.001** | - | **<0.001** | 0.13 | **<0.001** | **<0.001** |
| α-pinene | 0.15 | 0.99 | 0.98 | **0.02** | **0.03** | 0.18 | **<0.001** |
| Camphene | 0.57 | 0.88 | 0.35 | 0.56 | **0.01** | 0.55 | **0.01** |
| 3-carene | 0.43 | 0.90 | 0.92 | **0.01** | 0.29 | **0.05** | 0.36 |
| Limonene | 0.66 | 0.97 | 0.65 | **<0.001** | 0.46 | 0.59 | 0.40 |
| Eucalyptol | 0.39 | 0.86 | 0.61 | **0.004** | **0.01** | **<0.001** | 0.07 |
| Total BVOCs | **<0.001** | **<0.003** | 0.87 | **<0.001** | 0.23 | **0.01** | **0.003** |
| An | **<0.001** | **<0.001** | **0.001** | **<0.001** | **0.03** | **0.001** | **<0.001** |
| $G_S$ | **0.02** | 0.23 | 0.25 | **0.007** | 0.13 | **0.02** | **<0.001** |



**Table 2: The percentage of emissions that are dependent on light (PAR, in μmol m⁻² s⁻¹), as determined for the total monoterpene (MT) emission and for the main emitted compounds. The percentage was calculated as 100% × (light emissions - dark emissions)/light emissions. The numbers in brackets are the standard error of the mean. The trees that were measured were two individuals of English oak (*Quercus robur*), one European beech (*Fagus sylvatica*), two individuals of Norway spruce (*Picea abies*)**
5 **with an early budburst (Early spruce) and two individuals of Norway spruce with a late budburst (Late spruce). No data (n.d.) indicates compounds that were not detected in any sample or light level for that particular tree.**

| Tree | PAR | Total MT | Isoprene | α-pinene | Camphene | Sabinene | 3-Carene | Limonene | Eucalyptol |
|---|---|---|---|---|---|---|---|---|---|
| Oak 1 | 500 | 0 (0) | 100 (0) | 0 (0) | 17 (10) | n.d. | 0 (0) | 0 (0) | 0 (0) |
| | 1000 | 4 (4) | 100 (0) | 0 (0) | 17 (17) | n.d. | 11 (11) | 5 (5) | 0 (0) |
| | 1500 | 10 (10) | 100 (0) | 0 (0) | 40 (21) | n.d. | 0 (0) | 9 (9) | 3 (3) |
| Oak 2 | 500 | 0 (0) | 100 (0) | 0 (0) | 15 (10) | n.d. | 21 (21) | 0 (0) | 0 (0) |
| | 1000 | 15 (15) | 100 (0) | 16 (16) | 20 (20) | n.d. | 31 (18) | 15 (15) | 13 (13) |
| | 1500 | 0 (0) | 100 (0) | 12 (6) | 8 (8) | n.d. | 0 (0) | 0 (0) | 0 (0) |
| Beech | 500 | 6 (6) | n.d. | 0 (0) | 0 (0) | 100 (0) | 0 (0) | 0 (0) | 0 (0) |
| | 1000 | 23 (10) | n.d. | 20 (20) | 4 (4) | 100 (0) | 15 (15) | 0 (0) | 0 (0) |
| | 1500 | 52 (26) | n.d. | 7 (7) | 31 (31) | 100 (0) | 50 (6) | 77 (23) | 19 (19) |
| Early spruce 1 | 500 | 81 (5) | 100 (0) | 64 (9) | 6 (6) | n.d. | 88 (10) | 84 (1) | 89 (6) |
| | 1000 | 76 (6) | 100 (0) | 54 (8) | 10 (10) | n.d. | 79 (18) | 79 (1) | 89 (5) |
| | 1500 | 86 (3) | 100 (0) | 60 (8) | 14 (9) | n.d. | 73 (14) | 76 (3) | 91 (4) |
| Early spruce 2 | 500 | 18 (4) | 100 (0) | 20 (8) | 0 (0) | n.d. | 15 (15) | 18 (3) | 69 (3) |
| | 1000 | 0 (0) | 100 (0) | 3 (3) | 8 (8) | n.d. | 26 (26) | 0 (0) | 62 (4) |
| | 1500 | 19 (14) | 100 (0) | 43 (10) | 0 (0) | n.d. | 7 (7) | 0 (0) | 74 (3) |
| Late spruce 1 | 500 | 67 (14) | 100 (0) | 98 (2) | 12 (12) | n.d. | 100 (0) | 31 (25) | 95 (5) |
| | 1000 | 94 (3) | 100 (0) | 67 (33) | 45 (33) | n.d. | 100 (0) | 65 (32) | 100 (0) |
| | 1500 | 87 (3) | 100 (0) | 98 (2) | 0 (0) | n.d. | 100 (0) | 79 (16) | 100 (0) |
| Late spruce 2 | 500 | 26 (15) | 100 (0) | 85 (1) | 0 (0) | n.d. | 16 (13) | 8 (8) | 57 (22) |
| | 1000 | 68 (8) | 100 (0) | 91 (3) | 0 (0) | n.d. | 40 (8) | 37 (18) | 78 (14) |
| | 1500 | 57 (13) | 100 (0) | 85 (5) | 0 (0) | n.d. | 23 (12) | 20 (20) | 77 (15) |



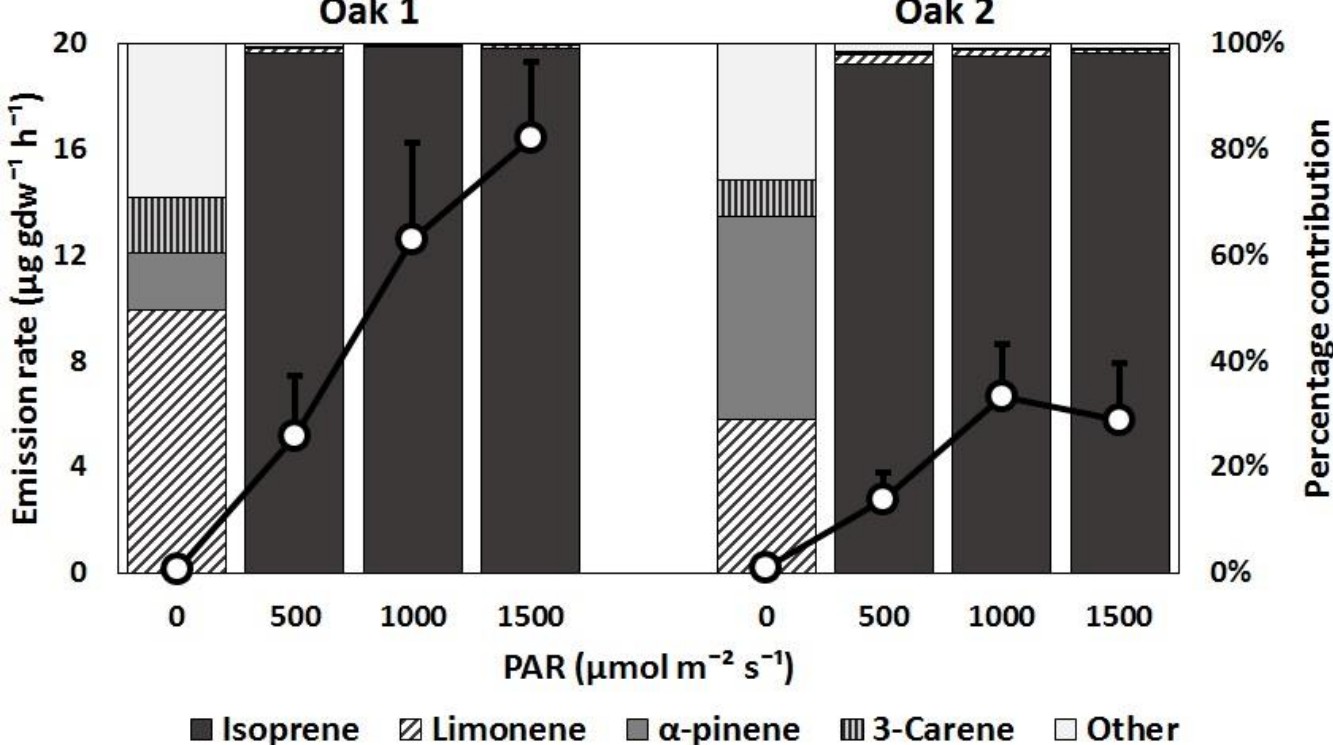

**Fig. 1: The total BVOC emission rate of two individual English oak trees (open circles) and the relative contribution of the major compounds at four intensities of photosynthetically active radiation (PAR). The error bars show the standard deviation, n = 3-5 leaves. The category "Other" contains the compounds tricyclene, camphene and eucalyptol.**





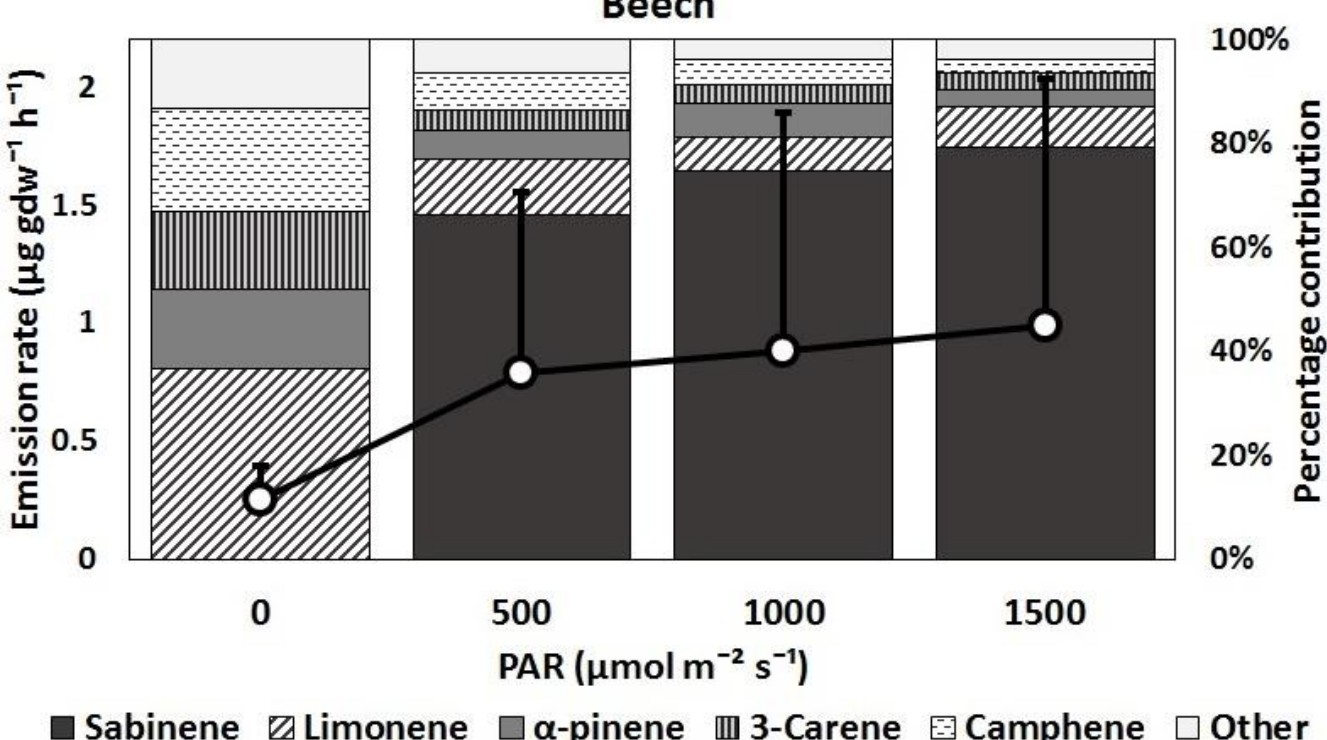

**Fig. 2: The total BVOC emission rate for European beech (open circles) and the relative contribution of the major compounds at four intensities of photosynthetically active radiation (PAR). The error bars show the standard deviation, n= 4-6 leaves. The category "Other" contains the compounds tricyclene, and eucalyptol.**





**Fig. 3: The total BVOC emission rate from two individuals of Norway spruce with an a) early budburst and with b) a late budburst and the relative contribution of the major compounds at four intensities of photosynthetically active radiation (PAR). The open circles show total monoterpene emission, whilst the open squares show isoprene emission of all measured twigs (n = 3-6 twigs). The error bars are the standard deviation of the data. The category "Other" contains the compounds tricyclene, β-pinene, eucalyptol and linalool for early spruce and tricyclene, β-pinene, α-terpinene eucalyptol and γ-terpinene for late spruce.**




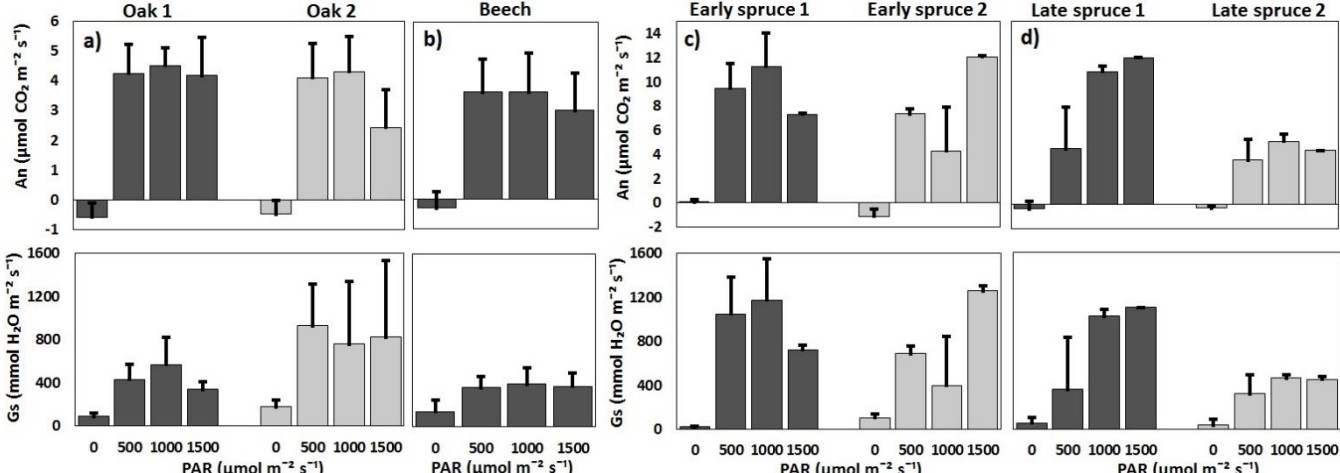

**Fig. 4:** The net assimilation rate (An) and stomatal conductance ($G_S$, mmol $H_2O$ m$^{-2}$ s$^{-1}$) of a) two individuals of English oak, b) European beech, c) two individuals of Norway spruce with an early budburst (Early spruce) and d) two individuals of Norway spruce with a late budburst (Late spruce). The values are averages ± the standard deviation (n = 13-21).