# Peer review of "Isoprenoid emission response to changing light conditions of English oak, European beech and Norway spruce"

_Biogeosciences, 2016_

## Referee Comment (RC1) · Anonymous Referee #1 · 30 Jan 2017

General comments:

This work is a nice study, which could add with no doubt one piece to this complicated puzzle of BVOC emission modeling improvement. However, there are still some inconsistencies and incoherence in the MS which make, at some points, its reading and understanding confusing. I recommend some major revisions of the MS (in its structure and interpretation, especially in the discussion part) before acceptation for publication.

Here are some leads which could help the authors.

The experimental strategy:

- Some important information in the sample strategy is missing (see also 'specific comments'), but briefly: we would like to understand better why these 3 species were selected? Why this kind of oak? How the two English oaks were selected; the 4 Norway spruces? How 'different' were they ... if they were? Were there genetically different? Why 'only' in July?

- Above all, why only the lowest (shaded) branches were sampled? I wonder how a light dependence behavior can be extrapolate from samples taken only from shaded branches, which, by definition, do not adapt very much to light? How could be sure that, some of the surprising discrepancies observed in the results, are not coming from the fact that shaded branches can present quite variable results?

- What about the weather conditions before the measurements? This can be of importance in the 'story' of the different replicate, and to understand some differences.

- In the last part of the MS, additional data from a (another?) 3 year experiment is mentioned: in addition to be not the right place to refer to this(these?) study(ies?), the reader cannot understand why all these date are not merged into one dataset?

The MS structure:

- Quite a lot of interesting points given in the 'discussion' section should be given in the 'results' one (see 'specific comments)

- The large discrepancies between the tree replicates is not enough highlighted in the 'results' section, especially for the Early spruce 2 which behaved very differently than other spruce replicates

- Why focusing/structuring the MS on a-pinene, camphene, 3-carene, limonene, eucalyptol? The reason that they were all produced by these species is not enough relevant; let's consider that a non a-pinene emitter were additionally studied, no focus would have been made on this - very - important MT! I'd rather be curious to learn about the total(MT), or the main compound emitted by each species.

- In the 'discussion' section, the MS wobbles constantly between 'compound division

point of view' and 'species division point of view': although I do not share the authors belief made for instance page 10 (l9-10) of a possible convergence of a light behavior for a same compound for different species (see further specific comments below), the authors has to decide: or they structure their MS on the compound basis or on the species dependency basis.

- Too many conclusions given in the discussion sound incoherent/inconsistent with what is discussed previously (see specific comments)

Specific comments:

- L27, p4: how the 'anticipated average daily temperature' was obtained?

- Section 3.5 title: could be shorter: 'Photosynthesis and stomatal conductance'

- L6, p8: why p>0.6?

- L18-19 p8 are not at the right place; it could be in the introduction of the discussion.

- L21-24, p8: rather at the end of the introduction/in the presentation of the strategy and objectives.

- Transition between l29 and l30 p8 is weird.

- Don't 'understand l6-7, p9 + the discussion is not on the same level (species effect) than just before (compound effect)

- I cannot figure out how a compound is going to be identically controlled by environmental conditions whatever the species. If so, it would make no sense to consider 'de novo' or 'pool' or 'de novo + pool' MT groups (cf. conclusion3-5 p9).

- L7-9, p9: I don't understand the last part of this sentence

- L9-11, p9: is a comparison between so different emitters than oak/beech and Norway spruce relevant?

- L12-21 p9 is more a 'result presentation' than a discussion; the early spruce 2 results

from Early spruce 2 are indeed different from the other; however the number of samples (n=13) is not that much different from the early spruce 1 (n=14) or oak 1(n=15) and cannot explain all these differences.

- Section from line 22 p9 has to been rewritten and better restructured : it starts from the strategy justification, jumps to camphene emission from different emitters (which was also discussed lines 1-5 of the same page 9), then to night emission (thus to the existence of storage structure) . . .

- Sentence l 3-5 p10 : I don't understand this sentence, since de novo emitting species ARE already considered as having a storage pool in emission models . . .

- L5-7 p10: few remarks:

o There is not so 'many' emission models available

o I don't see the link between the growing condition adaptions considered in some emission models and the experimental results presented in this study

o considering a compound division will not improve the plant functional division; it may (?) improve the emission model

- L9-10 p10: for decades, studies on BVOC emission variations showed that this 'promising idea' is not a good track to consider: a-pinene can be L-dependent for some emitter, but L-independent for other ones; I don't see how this would converge.

- section 4.2, p10: I think many results are here clearly presented in this 'discussion' section but should rather be added in the 'results' section which mainly (only?) presents light dependence results and not raw emissions (i.e., the main compounds measured, the ER values, their relative contribution, . . .)

- L22-23, p10: I'm getting confused: were these results obtained over a 3 year study?

- L24, p10: I'm not a native English speaker, but I would rather say something like 'the low variation in the emission pattern' rather than 'the low emission pattern variation'

- L27, p10: why considering now 'total emission' in a discussion section which states earlier that 'looking at emission patterns of separate compound would improve ...'; I cannot see the guiding thread of the discussion.

- L28-29 p10: I don't understand: even if some large differences exist between emission rates (ER) from different leaves, a light dependence (or not) can be study for each leave, whatever the absolute values of their emissions

- L 30-31, p10: stress and injury are different things + SQT are not the only compounds related to stress or injury

- L33, p10: what is 'the study' carried out in 2013?

- L1-2 p11: emission rates (ER) and standardized emissions (ER*) are not the same thing; top canopy ER are always 'much higher' than in the shaded canopy ER, but ER* are, or are not different. In this study only ER* were measured (30°, 1000 PAR).

- L6-7, p11: I don't understand the sentence

- L13, p11: a mysterious 3 year study is again mentioned; if these additional data are of importance they should be used and presented SINCE the beginning of the manuscript, not at this point of the 'discussion'.

- L18, p11: this conclusion seems incoherent with the points mentioned just before

- L19-22, p11: choosing shaded branches makes indeed the light dependency study over a large range of PAR not easy (possible?);

- L25, p11: another (last) mention of a 2014 study

Technical corrections

- Figures 1-3: please use colors rather than grey scales and above all, used the same color (or grey) for each compound in all the figures, otherwise it is quite difficult to follow

- Figure4: choose a color (or grey) for each tree or group of trees; or no color at all, but
not only 2 only different greys/colors for 4 different emitters

- I would not mind if the Appendice were a Table; in any case its presentation should be improved (e.g.: it is hard to understand which values correspond to which category '0' and '500' for oak 1, . . .)

---

## Referee Comment (RC2) · Anonymous Referee #2 · 13 Jun 2017

The light responses of BVOC emissions, photosynthesis and stomatal conductance on three common European tree species in Taastrup, Denmark were studied. Light intensity was increased in four steps, whilst other chamber conditions, temperature, humidity and $CO_2$ levels were fixed. The emission rate differed between individuals of the same species, the relative contributions of compounds to the total isoprenoid emission remained similar. Some compounds showed an increasing response with increasing light intensity, and other compounds no significant response. English oak and European beech showed high light-dependent emission fractions for isoprene and sabinene, but light-independent for other compounds. For the two provenances of Norway spruce, the compounds $\alpha$-pinene, 3-carene and eucalyptol showed high light-

dependent fractions. This study shows different responses of BVOC emissions to the change of light, which are useful for understanding BVOC emission characteristics, light dependent and light independent. It is suggested to be accepted for the publication after minor revision.

In Fig. 3: For early spruce 2, the emission rates of isoprene and total MT were extremely low at 1000 $\mu$mol m-2s-1 , please give an explanation.

It is suggested to develop relationships between BVOC emissions and PAR for these light dependant trees based on the measurements, or using current models to determine or valid relationships between BVOC emissions and PAR. Considering the valuable data, it is also suggested to publish it in more detail with this paper or somewhere else for the usage of the community.

---

## Author Response (AR1)

Author response to "Isoprenoid emission response to changing light conditions of English oak, European beech and Norway spruce"

Ylva van Meeningen, Guy Schurgers, Riikka Rinnan and Thomas Holst

5

30

**2017**

We thank the editor and the reviewers for their ideas and suggested improvements of this paper. All suggestions have been carefully considered in order to improve the readability of this manuscript. Below follows a list of changes made according

10 to each referee's suggestions.

**Reviewer #1**

We thank the reviewer for the thorough review of our manuscript and for the following suggested points of improvement.

Point 1: Some important information in the sample strategy is missing (see also 'specific comments'), but briefly: we would like to understand better why these 3 species were selected? Why this kind of oak? How the two English oaks were selected; the 4 Norway spruces? How 'different' were they... if they were? Were there genetically different? Why 'only' in July?

Thank you for your comment. A main advantage of using the IPG network is that all individuals of the available species used are genetically identical at all of the IPG sites. Each site in the network was initially provided with 2 individuals per species.
For the IPG network garden in Taastrup, we chose to do measurements on English oak, European beech and Norway spruce and we did measurements on all available trees that existed on this site. We chose these three species as they are common European tree species and considered to be high BVOC emitting trees, which we mention on P3 L28. Measurements were only carried out in July as this corresponded to the peak growing season with fully developed leaves and before autumn senescence. This was important to be able to assess responses to the different light levels. Information requested is mentioned in the beginning of the method section (P4 L5-12).

Point 2: Above all, why only the lowest (shaded) branches were sampled? I wonder how a light dependence behavior can be extrapolate from samples taken only from shaded branches, which, by definition, do not adapt very much to light? How could be sure that, some of the surprising discrepancies observed in the results, are not coming from the fact that shaded branches can present quite variable results?

We agree with the reviewer that shaded branches can present quite variable results in terms of emission pattern fluctuations. The decision to only measure the lowest positioned branches of the tree was based on an earlier study (Persson et al., 2016) performed on the same site on the same trees, where the emission patterns at different heights within the canopy were

35 investigated. The results showed little emission pattern difference between the upper and lower canopy levels for the oaks and the spruces, probably due to the relatively wide spacing in between the individual trees, which lets sunlight into the lower canopy as well. The only exception was the beech tree for which there was a clear difference. However, we cannot rule out that there have not been any impacts of the choice of using the lowest branches. This and the

However, we cannot rule out that there have not been any impacts of the choice of using the lowest branches. This and the possibly lower capacity to respond to increasing light availability in the shade-adapted leaves is mentioned in particular regarding the European beech tree the discussion section (P10 L32-33 and P11 L1-9). We also suggest doing more measurements in the upper canopy.

Point 3: What about the weather conditions before the measurements? This can be of importance in the 'story' of the different replicate, and to understand some differences.

Thank you for your comment. We agree that past weather events can influence the emission patterns of the studied tree species. The average weather conditions for July together with the ten-year average are mentioned in the method section (P4

- 5 L13-16). In the discussion section (P11 L23-26) we mention possible effect of drought on the emission patterns, which have been shown to have had an effect on one of the spruces, but we do not mention past weather conditions. We suggest adding a sentence regarding past weather events at P11 L27: "In comparison to the ten-year average weather conditions at the site, July in 2015 has had approximately the same amount of rainfall, but was approximately 2 °C colder. It is likely that the
- weather conditions might have had an effect on the emission results. However, as all trees have had the same exposure it 10 does not fully explain the different responses between trees."

Point 4: In the last part of the MS, additional data from a (another?) 3 year experiment is mentioned: in addition to be not the right place to refer to this(these?) study(ies?), the reader cannot understand why all these date are not merged into one dataset?

In the last part of the manuscript, we are referring to three different experiments which have been performed on the same site, but which have had different study settings and study aims. The results in 2013 was published in Persson et al. 2016, 2014 is unpublished measurement data in which light response were not studied and 2015 data are presented in the present 20 paper. The experiments are therefore not of the same design, but the intention of mentioning them is to point out that the compound composition is not only fairly similar between individuals within the same species, but over different years as well. We suggest rewriting the sentence on P11 L12 to the following: "This emission pattern difference between provenances have been observed for three separate studies performed at the same site (Persson et al. 2016; unpublished data in 2014 and current study in 2015)."

25

15

Point 5: The large discrepancies between the tree replicates is not enough highlighted in the 'results' section, especially for the Early spruce 2 which behaved very differently than other spruce replicates.

Thank you for your comment. We agree that for early spruce the differences in emissions between individuals could be mentioned in the text as well. We will therefore to add the emission values into the text and highlight these. 30

Point 6: Why focusing/structuring the MS on a-pinene, camphene, 3-carene, limonene, eucalyptol? The reason that they were all produced by these species is not enough relevant; let's consider that a non a-pinene emitter were additionally studied, no focus would have been made on this - very - important MT! I'd rather be curious to learn about the total(MT), or the main compound emitted by each species.

As we mention in our manuscript (P8 L27-29), we focus on these particular compounds because we wanted to look at the light responses of common compounds which we know are emitted from different tree species. So we took into consideration not only what compounds we found in our samples, but also which compounds were usually emitted from that

40 particular species. For example we did focus on sabinene as well, even though it was only emitted from European beech. Had it been that we had not had any a-pinene emission from one of the studied species, we would have considered it anyway as it is, as the reviewer points out, a very common and important MT.

We will add a sentence in 3.4 and 4.1 regarding the total MT emission response to light. Our suggestion is as follows (for P7 L15): "The light response for the total MT emission differed between species. Whilst the oaks and the second early spruce 45 showed little or no response to light, the beech and the remaining spruce trees increased their emissions."

For section 4.1, the light responses of MTs are already discussed for oak and beech (P9 L6-7). For spruce we suggest as follows (for P9 L12): "The two provenances of spruce responded differently with an increase in light, where the light dependent fraction of the total MT emission increased for all trees except for early spruce 2. Regarding separate compounds, they were also shown to respond differently with an increase in light depending on the individual tree."

35

Point 7: In the 'discussion' section, the MS wobbles constantly between 'compound division point of view' and 'species division point of view': although I do not share the authors belief made for instance page 10 (19-10) of a possible convergence of a light behavior for a same compound for different species (see further specific comments below), the authors has to decide: or they structure their MS on the compound basis or on the species dependency basis.

5

Thank you for your comment. As the tree species are quite different in their emission strategies, there are also going to be differences for some compounds in their emission responses. But we could also see that some compounds responded in a similar fashion despite the origin of the compound.

Our suggestion is to make a better species division. P8 L30-31 and P9 L1-5 could be removed from the article and rewritten into the article under the species where they are discussed in order to improve the readability of the text.

Point 8: L27, p4: how the 'anticipated average daily temperature' was obtained?

The anticipated average daily temperature was obtained by weighing in what the weather forecast has predicted for that particular day and based on personal experience regarding the site conditions, with the aim to create as stable and natural conditions as possible within the chamber.

Point 9: Section 3.5 title: could be shorter: 'Photosynthesis and stomatal conductance'

20 Thank you for your suggestion. We will revise the title according to the suggestion of the reviewer.

Point 10: *p8: why p>0.6?*

The p>0.6 refers to an a priori result which is not presented in Table 2, pointing out that there was no statistically significant increase in  $A_n$  with an increase in PAR.

Point 11: L18-19 p8 are not at the right place; it could be in the introduction of the discussion.

Thank you for your comment. We agree with this suggestion will move it to the end of the introduction of the discussion section (in old version this would be at P8 L24).

Point 12: L21-24, p8: rather at the end of the introduction/in the presentation of the strategy and objectives.

Thank you for your comment. We agree with this comment and move the sentence to the end of the introduction (P3 L27).

35

30

25

Point 13: Transition between l29 and l30 p8 is weird.

Thank you for your comment. Since P8 L26-29 is a repetition of the results, we will remove it from the MS to increase its readability.

40

Point 14: Don't 'understand l6-7, p9 + the discussion is not on the same level (species effect) than just before (compound effect)

We would argue that the discussion before the mentioned line numbers is a discussion that is both on a species and a compound level. Isoprene and camphene which are mentioned before all had similar responses to light independently of what species it was. Beech is an exception in regards to isoprene as it did not show a capacity to emit that particular compound. Oak and beech was combined in the above mentioned section as they also showed similar responses between species, both in regards to compounds and in regards to species effect. We will revise the manuscript to focus on the species effect in order to make the article a bit clearer.

Point 15: I cannot figure out how a compound is going to be identically controlled by environmental conditions whatever the species. If so, it would make no sense to consider 'de novo' or 'pool' or 'de novo + pool' MT groups (cf. conclusion3-5 p9).

10

5

15

As we argue a bit further down in the section (P10 L5-12), the results from this study are indicating that there might be similarities in emission responses between species. If that would be the case, it is true that the conception of what should be considered as de novo or pool groups need to be revised. But we are also arguing that the amount of data in this study is generally too low in order to make any robust conclusions. So yes, we can only argue that we saw an indication of similar

emission responses between species, but there is also a need to study it further.

Point 16: L7-9, p9: I don't understand the last part of this sentence.

- The argument of this sentence is that even though there has been an increasing amount of studies showing that de novo 20 species can have non-specific storing capacities (a suggestion is mentioned at P9 L34), there is still a tendency to consider some species as only light dependent or light independent. A suggestion is to rewrite the sentence at P9 L9-11 to the following: "For coniferous tree species, which are known to have storage structures contributing to a considerable lightindependent emission, a division of the emissions into light-dependent and light-independent fractions has been suggested 25 (Ghirardo et al., 2010). Although similar structures are absent in the broadleaf species studied here, the results suggest that
- these species also have a light-independent fraction."

Point 17: L9-11, p9: is a comparison between so different emitters than oak/beech and Norway spruce relevant?

30 It is true that the different emission strategies that exist between the three tree species could make it difficult to find any clear emission pattern similarities. We hope the suggested sentence in point 16 makes the comparison a bit clearer.

Point 18: L12-21 p9 is more a 'result presentation' than a discussion; the early spruce 2 results from Early spruce 2 are indeed different from the other; however the number of samples (n=13) is not that much different from the early spruce 1 (n=14) or oak 1(n=15) and cannot explain all these differences.

Thank you for your comment. We agree that the sample size for spruce 2 is not that much different from the sample sizes for the other studied trees and which is also mentioned on P10 L9-10, P12 L2-3 and P12 L16-17. We suggest to remove the sentence at P9 L12.

40

35

Point 19: Section from line 22 p9 has to been rewritten and better restructured: it starts from the strategy justification, jumps to camphene emission from different emitters (which was also discussed lines 1-5 of the same page 9), then to night emission (thus to the existence of storage structure)...

- 45 We can argue that the new section is not introduced as clearly as it could have been, jumping from individual compounds to discussing de novo and storage capacities. We suggest the following, starting from P9 L22: "Regarding the light dependency of MT emissions, there are several studies which have suggested that both de novo and storage pool emissions can occur within different tree species (Dindorf et al., 2006; Moukhtar et al., 2006; Ghirardo et al., 2010). Our study shows different compounds respond differently ... "We would also suggest to remove unnecessary repetition and to move the first three
- sentences (P9 L22-26) to the end of the section and adapt the first sentence in order to improve the readability. The following 50

is suggested, starting from the last sentence in the section: "These results suggest... de novo emitting tree species need to be considered to have storage pools in emission models as well. However, with the current experimental setup...".

Point 20: Sentence 1 3-5 p10: I don't understand this sentence, since de novo emitting species ARE already considered as 5 having a storage pool in emission models...

Unfortunately we cannot fully understand this point which the reviewer has given. If the reviewer wants to highlight that species can emit both light-dependently and light-independently in models, then that is true and we would rephrase the sentence. But if there is another opinion in the matter, we would need to ask the reviewer to clarify this further.

10

Point 21: L5-7 p10: few remarks:

o There is not so 'many' emission models available

o I don't see the link between the growing condition adaptions considered in some emission models and the experimental results presented in this study

o considering a compound division will not improve the plant functional division; it may (?) improve the emission model 15

Thank you for your comments, 'many' can indeed be removed from the manuscript. Regarding the second remark we would argue that the results in this study, where different emission amounts for different light levels are presented, could possibly help to improve the emission algorithms to be more accurate at different light levels. The last remark is correct and the sentence will be rephrased to "improve emission models" instead of "improve the division of plant functional types" (P10

20 L7).

Point 22: L9-10 p10: for decades, studies on BVOC emission variations showed that this 'promising idea' is not a good track to consider: a pinene can be L-dependent for some emitter, but L-independent for other ones: I don't see how this would converge.

25

45

We agree with the reviewer that for some compounds it has been shown that there has been both a light-dependent and lightindependent response depending on which species that has been considered. A-pinene is one of the more well studied compounds, due to its importance in atmospheric reactions and for being a commonly emitted compound by many different species. However, you could also argue that for other compounds, there is less information regarding their response and

which could also have an impact in different types of atmospheric reactions. We suggest to remove the sentence at P10 L9-10

Point 23: section 4.2, p10: I think many results are here clearly presented in this 'discussion' section but should rather be added in the 'results' section which mainly (only?) presents light dependence results and not raw emissions (i.e., the main 35 compounds measured, the ER values, their relative contribution, ...)

We will add the needed information in the results in condensed format and removing them from the discussion section.

Point 24: L22-23, p10: I'm getting confused: were these results obtained over a 3 year study? 40

As was mention in point 4, we are referring to previous measurements done on this same site, but which have had different experimental setups. What we wanted to emphasize is that our results are not only in line with other studies performed elsewhere, but that they are also consistent for the same trees over a number of years. We hope that the suggested revision mentioned in point 4 will correct this matter.

Point 25: L24, p10: I'm not a native English speaker, but I would rather say something like 'the low variation in the emission pattern' rather than 'the low emission pattern variation'

We would argue that the suggested change would refer to only one tree, whilst we want to stress the similarity between all of the trees. Our suggestion is as follows: "This would suggest that even if environmental factors such as temperature or light influence the total emission from oak, these do not alter the compound contribution to a great extent (Staudt et al., 2001; van Meeningen et al., 2016)."

Point 26: L27, p10: why considering now 'total emission' in a discussion section which states earlier that 'looking at emission patterns of separate compound would improve...'; I cannot see the guiding thread of the discussion.

Thank you for your comment. We suggest removing the sentence on P10 L27 and rewriting the sentence at P10 L27 to the following: "There were big differences in emission between beech leaves, making it difficult to see any clear increase in BVOC emission with an increase in light."

Point 27: L28-29 p10: I don't understand: even if some large differences exist between emission rates (ER) from different leaves, a light dependence (or not) can be study for each leave, whatever the absolute values of their emissions

15

It is true that the light dependence can be studied for each leaf separately. What we meant with the sentence was that the responses from the leaves were quite different, where some experienced an increase in their emissions with increasing light and where some were decreasing their emissions. We hope the above mentioned sentence in point 26 makes things clearer.

20 Point 28: L 30-31, p10: stress and injury are different things + SQT are not the only compounds related to stress or injury

As we are not focusing on stress or injury in this article, our suggestion is to remove the sentence to make a better flow in the text.

25 Point 29: L33, p10: what is 'the study' carried out in 2013?

The study which we refer to is a study performed in Taastrup in 2013 on this tree, but with a different experimental setup. It is mentioned in the section before (P10 L26), but of which we forgot to refer back to in this section. We will add in the reference into the sentence.

30

Point 30:  $L1-2 \ p11$ : emission rates (ER) and standardized emissions (ER\*) are not the same thing; top canopy ER are always 'much higher' than in the shaded canopy ER, but ER\* are, or are not different. In this study only ER\* were measured (30\_, 1000 PAR).

35 We agree that emission rates and standardized emission rates are not the same thing. We suggest to alter the emission rates into standardized emission rates in the sentence.

Point 31: L6-7, p11: I don't understand the sentence

40 Thank you for your comment. A suggestion is to remove the sentence as it is not providing much to the discussion.

Point 32: L13, p11: a mysterious 3 year study is again mentioned; if these additional data are of importance they should be used and presented SINCE the beginning of the manuscript, not at this point of the 'discussion'.

45 As mentioned in point 4 and 25, these are again not part of the same study but different studies with different experimental setups performed at the same trees. We hope the suggestion in point 4 makes the sentence more readable.

Point 33: L18, p11: this conclusion seems incoherent with the points mentioned just before

50 Our suggestion is to remove the sentence.

5

Point 34: L19-22, p11: choosing shaded branches makes indeed the light dependency study over a large range of PAR not easy (possible?);

- 5 As the reviewer points out, the response of shade adapted branches is most likely the same as if we would have done our measurements on sun adapted branches. As we point out on P11 L1-4, we have acknowledged that the European beech at this site had different standardized emission rates for the upper and lower parts of the canopy. However, for the remaining trees there was no statistically significant difference between the upper and lower levels, possibly due to wide spacing between trees which provides with light at all canopy levels of the tree. Even though it is uncertain what the response would
- 10 be at the upper canopy, we argued the adaptation at lower levels would still give us reasonable results. Furthermore, we would also argue that the response of the lowest positions branches should not be ignored, even though the response would most likely be quite different at the top of the canopy.

Point 35: Figures 1-3: please use colors rather than grey scales and above all, used the same color (or grey) for each compound in all the figures, otherwise it is quite difficult to follow

Thank you for your comment. We have chosen to have our figures in a grey scale as we did not want the reader to lose any information in case they decide to print the article in black and white. But we agree that for the case of sabinene for European beech and isoprene for English oak, they should be of different colors in order to emphasize that they are different compounds. Our suggestion is to change the color of isoprene to white, in order to be comparable with the Norway spruce

figures and keep the dark grey color for sabinene for the European beech tree.

Point 36: Figure4: choose a color (or grey) for each tree or group of trees; or no color at all, but not only 2 only different greys/colors for 4 different emitters

25

20

We will change the bars to be uniform in color.

Point 37: I would not mind if the Appendice were a Table; in any case its presentation should be improved (e.g.: it is hard to understand which values correspond to which category '0' and '500' for oak 1...)

30

We agree that the readability of the table can be improved, which we will change with the update of the manuscript.

**Reviewer #2**

We thank the reviewer for the thorough review of our manuscript and for the following suggested points of improvement.

35

Point 1: In Fig. 3: For early spruce 2, the emission rates of isoprene and total MT were extremely low at 1000 µmol m-2s-1, please give an explanation.

We were also surprised over the extremely low emission levels for early spruce 2 and can unfortunately not give any conclusive explanations. We believe though that the underlying reasons are most likely relatively high variation between samples, few samples to begin with and that the tree might not be fully recovered from a drought stress event which happened in 2013 (mentioned on P9 L16-18 and P11 L22-29).

On behalf of all authors,

45

Ylva van Meeningen

**Isoprenoid emission response to changing light conditions of English oak, European beech and Norway spruce – List of relevant changes**

- Introduction An additional sentence in the end of the section, which has been moved from the discussion section
- Results Additional emission values added and the 3.5 headline has been shortened
  - **Discussion** Values have been moved to the result, the sections have been revised and divided further into species, a few sentences on weather has been added
  - Appendix Has been divided into three sections instead of one
  - Figures Fig. 01 color for isoprene has been changed and Fig. 04 has a uniform colour

[revised manuscript text omitted]

|              |                   |                         |                     |                         |                             |                 |                             |                         |                             |                         |                             |                     | <del>0.01</del> |                 |                         |
|--------------|-------------------|-------------------------|---------------------|-------------------------|-----------------------------|-----------------|-----------------------------|-------------------------|-----------------------------|-------------------------|-----------------------------|---------------------|-----------------|-----------------|-------------------------|
| -1           | _                 | _                       | _                   | _                       | _                           | _               | _                           | _                       | _                           | _                       | _                           | _                   | _               | _               | _                       |
|              |                   |                         | <del><0.01</del> |                         |                             |                 |                             |                         |                             |                         |                             |                     |                 |                 |                         |
|              |                   |                         | ±                   | <del>0.02 ±</del>       | <del>0.02 ±</del>           |                 |                             | <del>0.01 ±</del>       |                             | <del>0.04 ±</del>       | <del>0.01 ±</del>           |                     |                 |                 | <del>0.10 ±</del>       |
|              | 0                 | <del>n.d.</del>         | <del><0.01</del> | <del><0.01</del>     | <del><0.01</del>         | <del>n.d.</del> | <del>n.d.</del>             | <del><0.01</del>     | <del>n.d.</del>             | 0.02                    | 0.01                        | <del>n.d.</del>     | <del>n.d.</del> | <del>n.d.</del> | 0.04                    |
| Forly        |                   |                         |                     |                         | <del><0.01</del>         |                 |                             |                         |                             |                         |                             |                     |                 |                 |                         |
| Contraction  |                   | <del>0.23 ±</del>       |                     | $0.02 \pm$              | ±                           |                 |                             | <del>0.01 ±</del>       |                             | <del>0.07_±</del>       | <del>0.04 ±</del>           |                     |                 |                 | 0.37 ±                  |
| spruce       | <del>500</del>    | <del><0.01</del>     | <del>n.d.</del>     | <del><0.01</del>     | <del><0.01</del>         | <del>n.d.</del> | <del>n.d.</del>             | <del><0.01</del>     | <del>n.d.</del>             | <del>0.01</del>         | <del><0.01</del>         | <del>n.d.</del>     | <del>n.d.</del> | <del>n.d.</del> | 0.02                    |
| ź            |                   |                         |                     | <del>0.01 ±</del>       | <del>0.01 ±</del>           |                 |                             | <del>0.01 ±</del>       |                             |                         | <del>0.02 ±</del>           |                     |                 |                 | <del>0.12 ±</del>       |
| (n=13        | + <del>1000</del> | 0.05                    | <del>n.d.</del>     | <del><0.01</del>     | 0.01                        | <del>n.d.</del> | <del>n.d.</del>             | <del><0.01</del>     | <del>n.d.</del>             | 0.01                    | 0.02                        | <del>n.d.</del>     | <del>n.d.</del> | <del>n.d.</del> | 0.13                    |
|              |                   |                         |                     |                         | <del><0.01</del>         |                 |                             |                         |                             |                         |                             |                     |                 |                 |                         |
|              |                   | <del>0.14 ±</del>       |                     | <del>0.03 ±</del>       | ±                           |                 | $0.02 \pm$                  | <del>0.01 ±</del>       |                             | <del>0.04 ±</del>       | <del>0.05 ±</del>           |                     |                 |                 | <del>0.29 ±</del>       |
|              | <del>1500</del>   | <del>0.05</del>         | <del>n.d.</del>     | <del><0.01</del>     | <del><0.01</del>         | <del>n.d.</del> | <del><0.01</del>         | <del><0.01</del>     | <del>n.d.</del>             | <del><0.01</del>     | <del><0.01</del>         | <del>n.d.</del>     | <del>n.d.</del> | <del>n.d.</del> | <del>0.05</del>         |
|              |                   |                         |                     |                         | <del>0.03 ±</del>           |                 |                             |                         |                             | <del>0.01 ±</del>       |                             |                     |                 |                 | <del>0.04 ±</del>       |
|              | 0                 | <del>n.d.</del>         | <del>n.d.</del>     | <del><0.01</del>     | 0.02                        | <del>n.d.</del> | <del>n.d.</del>             | <del>n.d.</del>         | <del>n.d.</del>             | <del>0.01</del>         | <del><0.01</del>         | <del>n.d.</del>     | <del>n.d.</del> | <del>n.d.</del> | <del>0.03</del>         |
| -            |                   | <del>0.31 ±</del>       |                     | $0.09 \pm$              | $\frac{0.02 \pm}{0.02 \pm}$ |                 |                             |                         |                             | $0.02 \pm$              | $\frac{0.01 \pm}{0.01}$     |                     |                 |                 | <del>0.49 ±</del>       |
| Late         | <del>500</del>    | <del>0.13</del>         | <del><0.01</del> | <del>0.09</del>         | <del>0.01</del>             | <del>n.d.</del> | <del>0.01</del>             | <del>0.01</del>         | <del>0.01</del>             | <del>0.03</del>         | <del>0.01</del>             | <del>n.d.</del>     | <del>n.d.</del> | <del>n.d.</del> | <del>0.30</del>         |
| spruce       | <del>)</del>      |                         | <del>0.01</del>     |                         |                             |                 |                             |                         |                             |                         |                             | <del>0.10</del>     |                 |                 |                         |
| +            | 4000              | <del>1.26 ±</del>       | ±                   | <del>0.29 ±</del>       | <del>0.21 ±</del>           |                 | <del>0.07 ±</del>           | $0.02 \pm$              | $0.04 \pm$                  | $0.05 \pm$              | $0.10 \pm$                  | ±                   |                 |                 | <del>2.16 ±</del>       |
| (n=13)       | +1000             | <del>0.49</del>         | <del>0.01</del>     | <del>0.26</del>         | <del>0.27</del>             | <del>n.d.</del> | <del>0.03</del>             | <del>0.01</del>         | <del>0.02</del>             | <del>0.05</del>         | <del>0.05</del>             | <del>0.06</del>     | <del>n.d.</del> | <del>n.d.</del> | <del>0.84</del>         |
|              |                   |                         | 0.01                |                         |                             |                 |                             |                         |                             |                         |                             | 0.03                |                 |                 |                         |
|              | 1500              | $0.54 \pm$              | ±                   | $0.23 \pm$              | $\frac{0.01 \pm}{0.01}$     |                 | $\frac{0.03 \pm}{0.03 \pm}$ | $0.01 \pm$              | $\frac{0.02 \pm}{0.02 \pm}$ | $0.04 \pm$              | $\frac{0.03 \pm}{0.03 \pm}$ | ±                   |                 |                 | 0.93 ±                  |
|              | <del>1500</del>   | 0.04                    | <del><0.01</del> | <del>0.01</del>         | <del><0.01</del>         | <del>n.d.</del> | <del><0.01</del>         | <del><0.01</del>     | <del><0.01</del>         | <del><0.01</del>     | <del><0.01</del>         | <del><0.01</del> | <del>n.d.</del> | <del>n.d.</del> | 0.04                    |
| -            | -                 | -                       | -                   | -                       | -                           |                 | -                           | -                       | -                           | -                       | -                           | -                   | -               | -               | -                       |
|              | 0                 |                         |                     | $\frac{0.02 \pm}{0.01}$ | $\frac{0.03 \pm}{0.02}$     |                 |                             | $\frac{0.04 \pm}{0.01}$ |                             | $0.04 \pm$              | $\frac{0.01 \pm}{0.01}$     |                     |                 |                 | $0.25 \pm$              |
|              | <del>Q</del>      | <del>n.d.</del>         | <del>n.d.</del>     | 0.01                    | 0.02                        | <del>n.d.</del> | <del>n.d.</del>             | 0.01                    | <del>n.d.</del>             | 0.02                    | 0.01                        | <del>n.d.</del>     | n.d.            | <del>n.d.</del> | 0.12                    |
|              |                   | 0.20                    |                     | 0.14                    | <del><0.01</del>         |                 | 0.00                        | 0.05                    |                             | 0.06                    | 0.00                        |                     |                 |                 | 1.17                    |
| Late         | 500               | $\frac{0.30 \pm}{0.16}$ |                     | $0.14 \pm$              | ±                           |                 | $\frac{0.02 \pm}{0.02}$     | $0.05 \pm$              |                             | 0.06 ±                  | $\frac{0.02 \pm}{0.02}$     |                     |                 |                 | 1.1/±                   |
| spruce       | <del>, 900</del>  | 0.16                    | <del>n.d.</del>     | 0.07                    | <del><0.01</del>         | <del>n.d.</del> | 0.02                        | <del>0.04</del>         | <del>n.d.</del>             | 0.05                    | 0.02                        | <del>n.d.</del>     | <del>n.d.</del> | <del>n.d.</del> | 0.70                    |
| 2 |                   | 0.45                    |                     | 0.00                    | <del><0.01</del>         |                 | 0.00                        | 0.04                    |                             | 0.04                    | 0.00                        | 0.02                |                 |                 | 1.00                    |
| (n=18)       | + 1000            | $\frac{0.45 \pm}{0.00}$ |                     | $\frac{0.20 \pm}{0.01}$ | ±                           |                 | $\frac{0.03 \pm}{0.01}$     | $\frac{0.06 \pm}{0.01}$ |                             | $\frac{0.06 \pm}{0.01}$ | $\frac{0.03 \pm}{0.01}$     | • <del>*</del>      |                 |                 | $\frac{1.69 \pm}{0.12}$ |
|       | × <del>1000</del> | 0.00                    | <del>n.a.</del>     | 0.01                    | <del><0.01</del>         | <del>n.a.</del> | <del><0.01</del>         | <del><0.01</del>     | <del>n.a.</del>             | 0.01                    | <del><0.01</del>         | 0.02                | <del>n.a.</del> | <del>n.a.</del> | 0.12                    |
|              |                   | 0.22                    |                     | 0.015                   | <del><0.01</del>         |                 | 0.02                        | 0.05                    |                             | 0.05                    | 0.04                        | 0.04                |                 |                 | 1.41                    |
|              | 1500              | 0.02                    | n d                 | 0.013                   | -0.01                       | n d             | 0.03 ±                      | 0.01                    | d                           | 0.01                    | 0.04 ±
<0.01             | -0.01               | d               | n d             | 1./11 ±                 |
|              | 1500              | 0.05                    | <del>n.a.</del>     | $\pm 0.04$              | <del><0.01</del>         | <del>n.a.</del> | 0.01                        | 0.01                    | <del>n.a.</del>             | 0.01                    | <del><0.01</del>         | <del><0.01</del> | <del>n.a.</del> | <del>n.a.</del> | <del>0.08</del>         |

Table A1. The mean average actual emission ( $\pm$  standard deviation) of detected compounds at light levels (PAR) 0, 500, 1000 and 1500 µmol m-2 s-1 and the number of samples taken from English oak (*Quercus robur*) in µg gdw-1 h-1. No data (n.d.) indicates that the compound was not detected in any sample for the measured light level on that particular tree.

| Tree            |                              | Oak 1 (              | n=15)                    |                           | Oak 2 (n=17)               |                           |                           |                               |  |  |
|-----------------|------------------------------|-----------------------------|---------------------------------|---------------------------|-----------------------------------|---------------------------|---------------------------|-------------------------------|--|--|
| PAR      | 0                     | 500                  | 1000                     | 1500               | 0                          | 500                | 1000               | 1500                   |  |  |
| ISO      | n.d.                  | $5.08 \pm 2.32$             | $12.53 \pm 3.68$                | $16.31 \pm 2.91$          | 0.02                              | $2.68 \pm 0.99$           | $6.53 \pm 2.0$            | 5.68 ± 2.22            |  |  |
| Tricyclene      | <0.01              | $\leq 0.01 \pm < 0.01$      | $\underline{0.01 \pm {<} 0.01}$ | $0.01 \pm < 0.01$         | <0.01                   | <0.01           | 0.01               | 0.01                   |  |  |
| α-pinene | $\underline{0.02 \pm <0.01}$ | $0.01 \pm 0.01$             | $\underline{0.01 \pm {<}0.01}$  | $0.01 \pm < 0.01$         | $\underline{0.01 \pm <\!\! 0.01}$ | $\underline{0.01\pm0.01}$ | $\underline{0.02\pm0.01}$ | $\underline{0.03 \pm <} 0.01$ |  |  |
| Camphene        | $\underline{0.03\pm0.02}$    | $\underline{0.01 \pm 0.01}$ | $\underline{0.01\pm0.01}$       | $\underline{0.02\pm0.01}$ | $0.01 \pm < 0.01$                 | $\underline{0.04\pm0.02}$ | $\underline{0.05\pm0.02}$ | $\underline{0.04\pm0.03}$     |  |  |
| Sabinene        | n.d.                  | n.d.                 | n.d.                     | n.d.               | n.d.                       | n.d.               | n.d.               | n.d.                   |  |  |
| β-pinene        | n.d.                  | n.d.                 | n.d.                     | n.d.               | n.d.                       | n.d.               | n.d.               | n.d.                   |  |  |
| 3-Carene        | $0.02 \pm < 0.01$            | $0.01 \pm < 0.01$           | $\underline{0.01\pm0.01}$       | $0.01 \pm < 0.01$         | $0.01 \pm < 0.01$                 | $\underline{0.01\pm0.01}$ | $0.01 \pm < 0.01$         | $\underline{0.01 \pm 0.01}$   |  |  |

| a-terpinene | n.d.               | n.d.               | n.d.                   | n.d.                 | n.d.                 | n.d.                 | n.d.                 | n.d.                 |
|--------------------|---------------------------|---------------------------|-------------------------------|-----------------------------|-----------------------------|-----------------------------|-----------------------------|-----------------------------|
| Limonene           | $\underline{0.08\pm0.04}$ | $0.05 \pm 0.03$           | $\underline{0.05\pm0.03}$     | $\underline{0.06 \pm 0.03}$ | $\underline{0.03 \pm 0.01}$ | $\underline{0.08 \pm 0.04}$ | $\underline{0.09 \pm 0.02}$ | $\underline{0.08 \pm 0.01}$ |
| Eucalyptol         | $0.02 \pm < 0.01$         | $\underline{0.01\pm0.01}$ | $\underline{0.01\pm {<}0.01}$ | $0.01 \pm < 0.01$           | $0.01 \pm < 0.01$           | $\underline{0.02\pm0.01}$   | $0.02 \pm < 0.01$           | $0.03 \pm < 0.01$           |
| y-terpinene | n.d.               | n.d.               | n.d.                   | n.d.                 | n.d.                 | n.d.                 | n.d.                 | n.d.                 |
| Linalool           | n.d.               | n.d.               | n.d.                   | n.d.                 | n.d.                 | n.d.                 | n.d.                 | n.d.                 |
| SQT         | n.d.               | n.d.               | n.d.                   | n.d.                 | n.d.                 | n.d.                 | n.d.                 | n.d.                 |
| Total              | $0.16 \pm 0.08$           | 5.19 ± 2.27        | 12.62 ± 3.65           | $16.43 \pm 2.86$            | $0.05 \pm 0.03$             | 2.79 ± 1.01          | 6.68 ± 1.95          | 5.79 ± 2.14          |

Table A2. The mean average actual emission ( $\pm$  standard deviation) of detected compounds at light levels (PAR) 0, 500, 1000 and 1500 µmol m-2 s-1 and the number of samples taken from European beech (*Fagus sylvatica*) in µg gdw-1 h-1. No data (n.d.) indicates that the compound was not detected in any sample for the measured light level on that particular tree.

| Tree               |                           | Beech              | ( n=21)     |                    |
|--------------------|---------------------------|--------------------|--------------------|--------------------|
| PAR                | 0                  | 500         | 1000        | 1500        |
| ISO                | n.d.               | n.d.        | n.d.        | n.d.        |
| Tricyclene         | n.d.               | 0.01 ± 0.02 | 0.01        | 0.01        |
| a-pinene    | $0.04 \pm 0.03$           | $0.04 \pm 0.02$    | $0.06 \pm 0.06$    | $0.03 \pm 0.03$    |
| Camphene           | $0.05 \pm 0.03$           | $0.06 \pm 0.03$    | $0.04 \pm 0.02$    | $0.03 \pm 0.02$    |
| Sabinene           | n.d.               | $0.52 \pm 0.78$    | 0.65 ± 0.97 | 0.75 ± 1.05 |
| β-pinene    | n.d.               | n.d.        | n.d.        | n.d.        |
| 3-Carene    | $0.04 \pm 0.03$           | $0.03 \pm 0.02$    | 0.03 ± 0.03 | $0.03 \pm 0.02$    |
| a-terpinene | n.d.               | n.d.        | n.d.        | n.d.        |
| Limonene           | $0.09 \pm 0.05$           | $0.09 \pm 0.07$    | $0.06 \pm 0.05$    | 0.07 ± 0.03 |
| Eucalyptol  | $0.03 \pm 0.01$           | $0.04 \pm 0.01$    | 0.03 ± 0.01 | 0.03 ± 0.01 |
| y-terpinene | n.d.               | n.d.        | n.d.        | n.d.        |
| Linalool           | n.d.               | n.d.        | n.d.        | n.d.        |
| SQT         | n.d.               | n.d.        | 0.02               | $0.04 \pm 0.06$    |
| Total       | $\underline{0.25\pm0.14}$ | $0.79 \pm 0.76$    | $1.23 \pm 1.18$    | 0.99 ± 1.05 |

5

Table A3. The mean average actual emission (± standard deviation) of detected compounds at light levels (PAR) 0, 500, 1000 and 1500 µmol m-2 s-1 and the number of samples taken from the two provenances of spruce (*Picea abies*) in µg gdw-1 h-1. No data (n.d.) indicates that the compound was not detected in any sample for the measured light level on that particular tree.

[revised manuscript text omitted]

I

---

## Referee Report (RR1)

Review revision of Isoprenoid emission response to changing light conditions of English oak, European beech and Norway spruce

The author answered my first question. But, it is a pity that no relationships between BVOC emissions and PAR for these light dependant trees were investigated. It is better to do that in the future using a lot of observational data.
I suggest it can be accepted.